# A quadratic model captures the human V1 response to variations in chromatic direction and contrast

Michael A Barnett[1]*, Geoffrey K Aguirre[2], David Brainard[1]

[1]Department of Psychology, University of Pennsylvania, Philadelphia, United States;
[2]Department of Neurology, University of Pennsylvania, Philadelphia, United States

**Abstract** An important goal for vision science is to develop quantitative models of the representation of visual signals at post-receptoral sites. To this end, we develop the quadratic color model (QCM) and examine its ability to account for the BOLD fMRI response in human V1 to spatially uniform, temporal chromatic modulations that systematically vary in chromatic direction and contrast. We find that the QCM explains the same, cross-validated variance as a conventional general linear model, with far fewer free parameters. The QCM generalizes to allow prediction of V1 responses to a large range of modulations. We replicate the results for each subject and find good agreement across both replications and subjects. We find that within the LM cone contrast plane, V1 is most sensitive to L-M contrast modulations and least sensitive to L+M contrast modulations. Within V1, we observe little to no change in chromatic sensitivity as a function of eccentricity.

**\*For correspondence:**
micalan@sas.upenn.edu

**Competing interests:** The authors declare that no competing interests exist.

## Introduction

The initial stage of human color vision is well characterized. The encoding of light by the three classes of cone photoreceptors (L, M, and S) is described quantitatively by a set of spectral sensitivity functions, one for each class. Knowledge of the spectral sensitivities allows for the calculation of cone excitations from the spectral radiance of the light entering the eye (*Brainard and Stockman, 2010*). This quantitative characterization supports the analysis of the information available to subsequent processing stages (*Geisler, 1989*; *Cottaris et al., 2019*), supports the precise specification of visual stimuli (*Brainard, 1996*; *Brainard et al., 2002*), and enables color reproduction technologies (*Wandell and Silverstein, 2003*; *Hunt, 2004*). An important goal for vision science is to develop similarly quantitative models for the representation of visual signals at post-receptoral sites.

The second stage of color vision combines the signals from the cones to create three post-receptoral mechanisms. Psychophysical evidence supports the existence of two cone-opponent mechanisms, which represent differences between cone signals (S-(L+M) and L-M), and a luminance mechanism, which represents an additive combination (L+M) (*Krauskopf et al., 1982*; *Stockman and Brainard, 2010*). Physiological evidence shows that this recombination begins in the retina with correlates observed in the responses of retinal ganglion cells and subsequently in the neurons of the lateral geniculate nucleus (*De Valois et al., 1966*; *Derrington et al., 1984*; *Lennie and Movshon, 2005*). While the outlines of this second stage seem well established, the precise links between retinal physiology and visual perception remain qualitative and subject to debate (*Stockman and Brainard, 2010*; *Shevell and Martin, 2017*).

Studies focused on developing quantitative parametric models of the chromatic response properties of neurons in primary visual cortex of primates (area V1) have not yet converged on a widely accepted model (*Johnson et al., 2004*; *Solomon and Lennie, 2005*; *Tailby et al., 2008*; *Horwitz and Hass, 2012*; *Weller and Horwitz, 2018*). In part, this is due to the considerable

heterogeneity of chromatic response properties found across individual cortical neurons (*Gegenfurtner, 2001*; *Lennie and Movshon, 2005*; *Solomon and Lennie, 2007*; *Shapley and Hawken, 2011*; *Horwitz, 2020*). In addition, variation in stimulus properties across studies limits the ability to compare and integrate results.

The chromatic response of V1 has also been studied using blood oxygen level dependent (BOLD) functional magnetic resonance imaging (fMRI) (*Wandell et al., 2006*). This includes studies that characterize the relative responsiveness of V1 (and other visual areas) to various chromatic and achromatic stimuli (*Engel et al., 1997*; *Hadjikhani et al., 1998*; *Beauchamp et al., 1999*; *Bartels and Zeki, 2000*; *Mullen et al., 2007*; *Goddard et al., 2011*; *Lafer-Sousa et al., 2016*) and how this depends on the spatial and temporal properties of the stimulus (*Liu and Wandell, 2005*; *D'Souza et al., 2016*; *Mullen et al., 2010b*).

Few studies, however, have pursued a quantitative model of the V1 BOLD response to arbitrary chromatic stimulus modulations. Development of such a model is important, since it would enable generalizations of what is known from laboratory measurements to natural viewing environments, where stimuli rarely isolate single mechanisms. Further, the parameters of such a model provide a succinct summary of processing that could be used to understand the flow of chromatic information through cortex. Notably, *Engel et al., 1997* conducted a pioneering study that varied the chromatic content and temporal frequency of stimuli and observed that the V1 BOLD fMRI signal was maximally sensitive to L-M stimulus modulations.

In the present study, we focus on the signals that reach V1 from stimulus modulations confined to the L- and M-cone contrast plane (LM contrast plane). Specifically, we measured responses with fMRI to flickering modulations designed to systematically vary combinations of L- and M-cone contrast. Using these data, we developed a model—the quadratic color model (QCM)—that predicts the V1 BOLD fMRI response for any arbitrary stimulus in the LM contrast plane, using a small set of parameters. We validate the QCM through comparison to a less constrained general linear model (GLM). Importantly, the parameters of the QCM are biologically meaningful, and describe the sensitivity of V1 to chromatic modulations. Further, we generate cortical surface maps of model parameters across early visual cortex, allowing us to examine how chromatic sensitivity changes across V1 as a function of visual field eccentricity.

## Quadratic color model (QCM)

This section provides an overview of the Quadratic Color Model (QCM); a full mathematical description is provided in the Appendix 1. Given a description of the stimulus, the QCM provides a prediction of the BOLD fMRI response within V1. Our stimuli were full field temporal chromatic modulations that can be specified by their contrast (vector length of the stimulus in the LM contrast plane) and chromatic direction (angle of the stimulus measured counterclockwise with respect to the positive abscissa). From this stimulus specification, the model employs three stages that convert the input to the BOLD fMRI response (*Figure 1*). First, a quadratic isoresponse contour is defined that allows for the transformation of contrast and direction into what we term the 'equivalent contrast'. Second, a single non-linear function transforms the equivalent contrast to a prediction of the population neuronal response underlying the BOLD response. Finally, the neuronal response is converted to a predicted BOLD response by convolution with a hemodynamic response function.

### Isoresponse contours and equivalent contrast

The first stage of the QCM computes the equivalent contrast of a stimulus from its cone contrast using a subject-specific elliptical isoresponse contour. Equivalent contrast is the effective contrast of a stimulus in V1 once it has been adjusted to account for differences in the neuronal sensitivity to stimulation across different chromatic directions. An isoresponse contour is defined as a set of stimuli that evoke the same neuronal response. In the QCM, the loci of such stimuli form an elliptical isoresponse contour in the LM cone contrast plane. All points on this elliptical isoresponse contour have the same equivalent contrast (*Figure 1A*, dashed gray ellipses). As the amplitude of the neuronal response increases, the ellipse that defines the set of stimuli producing that response also grows in overall scale. Importantly, the QCM assumes that the aspect ratio and orientation of elliptical isoresponse contours do not change as a function of the response level; only the overall scale of the ellipse changes. The use of elliptical isoresponse contours is motivated by prior psychophysical

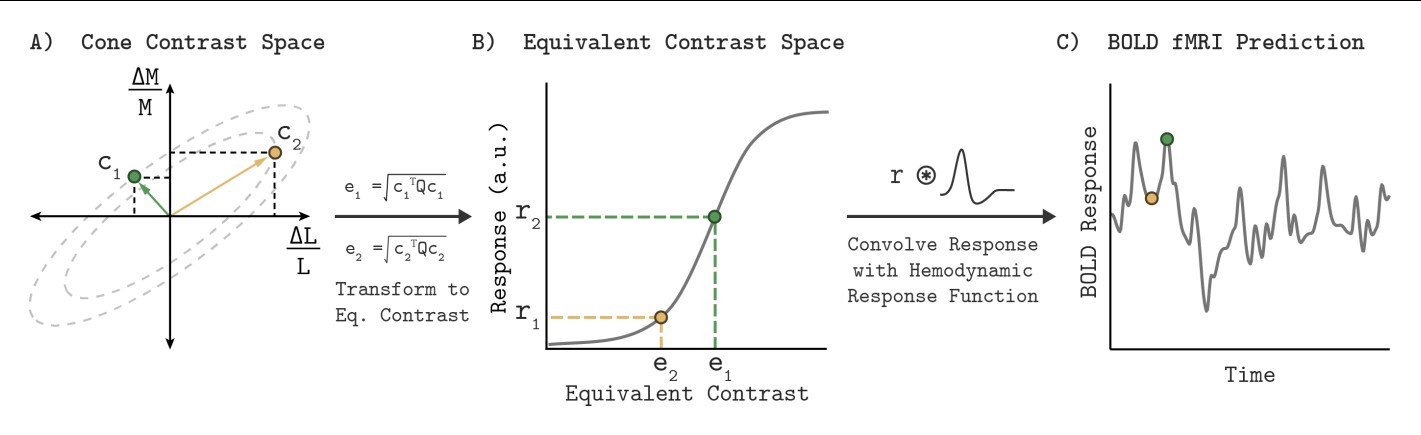

**Figure 1.** Quadratic color model. (**A**) The LM contrast plane representing two example stimuli ($c_1$ and $c_2$) as the green and yellow vectors. The vector length and direction specify the contrast and chromatic direction of the positive arm of the symmetric modulation (see Visual Stimuli in Materials and methods). Using the parameters of an elliptical isoresponse contour (panel A, dashed gray ellipses), fit per subject, we can construct a 2x2 matrix Q that allows us to compute the equivalent contrast of any stimulus in the LM contrast plane (panel B; $e_1$ and $e_2$; see Appendix 1). (**B**) Transformation of equivalent contrast to neuronal response. The equivalent contrasts of the two example stimuli from panel A are plotted against their associated neuronal response. A single Naka-Rushton function describes the relationship between equivalent contrast and the underlying neuronal response. (**C**) To predict the BOLD fMRI response, we convolve the neuronal response output of the Naka-Rushton function with a subject-specific hemodynamic response function. Note that the BOLD fMRI response prediction for the green point is greater than the prediction for the yellow point, even though the yellow point has greater cone contrast. This is because of where the stimuli lie relative to the isoresponse contours. The difference in chromatic direction results in the green point producing a greater equivalent contrast, resulting in the larger BOLD response.

(*Poirson et al., 1990*; *Knoblauch and Maloney, 1996*), electrophysiological (*Horwitz and Hass, 2012*), and fMRI experiments (*Engel et al., 1997*) which have successfully used ellipses to model chromatic isoresponse contours.

The elliptical isoresponse contours are described by a symmetric quadratic function that defines the major and minor axes of the ellipse. We use this quadratic function to compute the equivalent contrast for each stimulus. The vector lengths of all stimuli that lie on a single isoresponse contour provide the cone contrasts required to elicit an equal neuronal response. The minor axis of the elliptical isoresponse contour corresponds to the chromatic direction that requires the least amount of cone contrast to produce this equal neuronal response, and is therefore the most sensitive chromatic direction. The major axis corresponds to the direction of least sensitivity. At this stage, the model is only concerned with the shape of the elliptical contour, thus we adopt the convention of normalizing the ellipse used to define equivalent contrast so that its major axis has unit length. This allows the length of the minor axis to directly represent the relative sensitivity, which is taken as a ratio of the minor axis (maximal sensitivity) to major axis (minimal sensitivity), referred to as the minor axis ratio. The angle of the major axis in the LM contrast plane (ellipse angle) orients these maximally and minimally sensitive directions.

## Response non-linearity

Since all of the stimuli that lie on a single isoresponse contour produce the same response, we can represent these points by their common equivalent contrast. The neuronal responses to stimuli across different color directions are a function of this single variable, and therefore we can transform equivalent contrast into predicted neuronal response via a single static non-linear function (*Figure 1B*). Here, we employ the four-parameter Naka-Rushton function (see Appendix 1).

## Transformation to BOLD fMRI signal

To predict the BOLD fMRI signal, we obtain the time-varying neuronal response prediction from the Naka-Rushton function for a stimulus sequence presented in the fMRI experiment. This neuronal response is convolved with a subject-specific hemodynamic response function to produce a prediction of the BOLD fMRI signal (*Figure 1C*).

## QCM summary

In summary, the QCM takes as input the temporal sequence of stimulus modulations, defined by their chromatic direction and contrast in the LM cone contrast plane, and outputs a prediction of the BOLD fMRI time course. The QCM has six free parameters: two that define the shape of the normalized elliptical isoresponse contour and four that define the Naka-Rushton equivalent contrast response function.

## Results

To evaluate the QCM, three subjects underwent fMRI scanning while viewing stimuli consisting of spatially uniform (0 cycles per degree) chromatic temporal modulations, presented using a block design. Each 12 s block consisted of a 12 Hz bipolar temporal modulation in one of 8 chromatic directions and at one of 5 log-spaced contrast levels. We split the chromatic directions into two sessions and subjects viewed each of the 20 combinations of chromatic direction (four directions) and contrast (five levels) once per run in a pseudorandomized order (see Materials and methods, Figure 10 and 11). For each subject, a measurement set consisted of 20 functional runs conducted across the two scanning sessions. We collected two complete measurement sets (referred to as Measurement Set 1 and 2) for each subject, and fit the model to each set separately to test for the replicability of our findings. We first modeled the data using a conventional GLM that accounts for the response to each of the 40 stimulus modulations independently. The fit of this relatively unconstrained model was used as a benchmark to evaluate the performance of QCM. Results were similar for all three subjects. In the main text, we illustrate our findings with the data from one subject (Subject 2); results from the other two subjects may be found in the supplementary materials.

## Characterizing cortical responses with a conventional GLM

### Contrast-response functions

To examine the basic chromatic response properties of V1, we grouped the GLM beta weights by their corresponding chromatic direction and plotted them as a function of contrast, indicated as the filled circles in each of the eight panels of *Figure 2* (data from Subject 2). For each chromatic direction, the V1 BOLD response generally increased with contrast, as expected. This result is consistent across the two independent measurement sets for Subject 2, as can be seen by comparing the green and purple points in *Figure 2*. Further, the increasing response with stimulus contrast was also observed in both measurement sets for the other two subjects (*Figure 2—figure supplements 1–2*).

The rate at which V1 BOLD responses increase with contrast varied with chromatic direction. This can be seen in *Figure 2* by noting that the maximum stimulus contrast differed considerably across chromatic directions, while the maximum response remained similar. For example, a modulation in the 45° direction required ~60% stimulus contrast to elicit a response of 0.6 while stimuli modulated in the −45° direction required only ~12% stimulus contrast to produce a similar response.

The GLM places no constraints on the values of GLM beta weights, and we observed that these values did not always increase monotonically with contrast. Given the a priori expectation that the BOLD response itself increases monotonically with contrast, this raises the possibility that the GLM overfits the data, using its flexibility to account for the noise as well as the signal in the response. To examine this, we fit a series of more constrained models that enforce the requirement that the fitted response within chromatic direction increases monotonically with contrast. These models employed a Naka-Rushton function to describe the contrast response function in each chromatic direction. Across the models, we constrained varying numbers of the parameters to be constant across chromatic direction. The most general of these models fits a separate Naka-Rushton function to each color direction, allowing all but the offset parameter to be independent across chromatic directions. We also explored locking the amplitude parameter (in addition to the offset), the exponent parameter (in addition to the offset), and the amplitude, exponent, and offset parameters (allowing only the semi-saturation parameter to vary with chromatic direction). To evaluate how well these models fit the data, we ran a cross-validation procedure, described below, to compare the Naka-Rushton model fits with those of the GLM. The cross-validated $R^2$ for all of the Naka-Rushton models was slightly better than for the GLM, indicating that enforcing smooth monotonicity reduces a slight overfitting. These cross-validation results can be seen in *Figure 4—figure supplement 1*. For

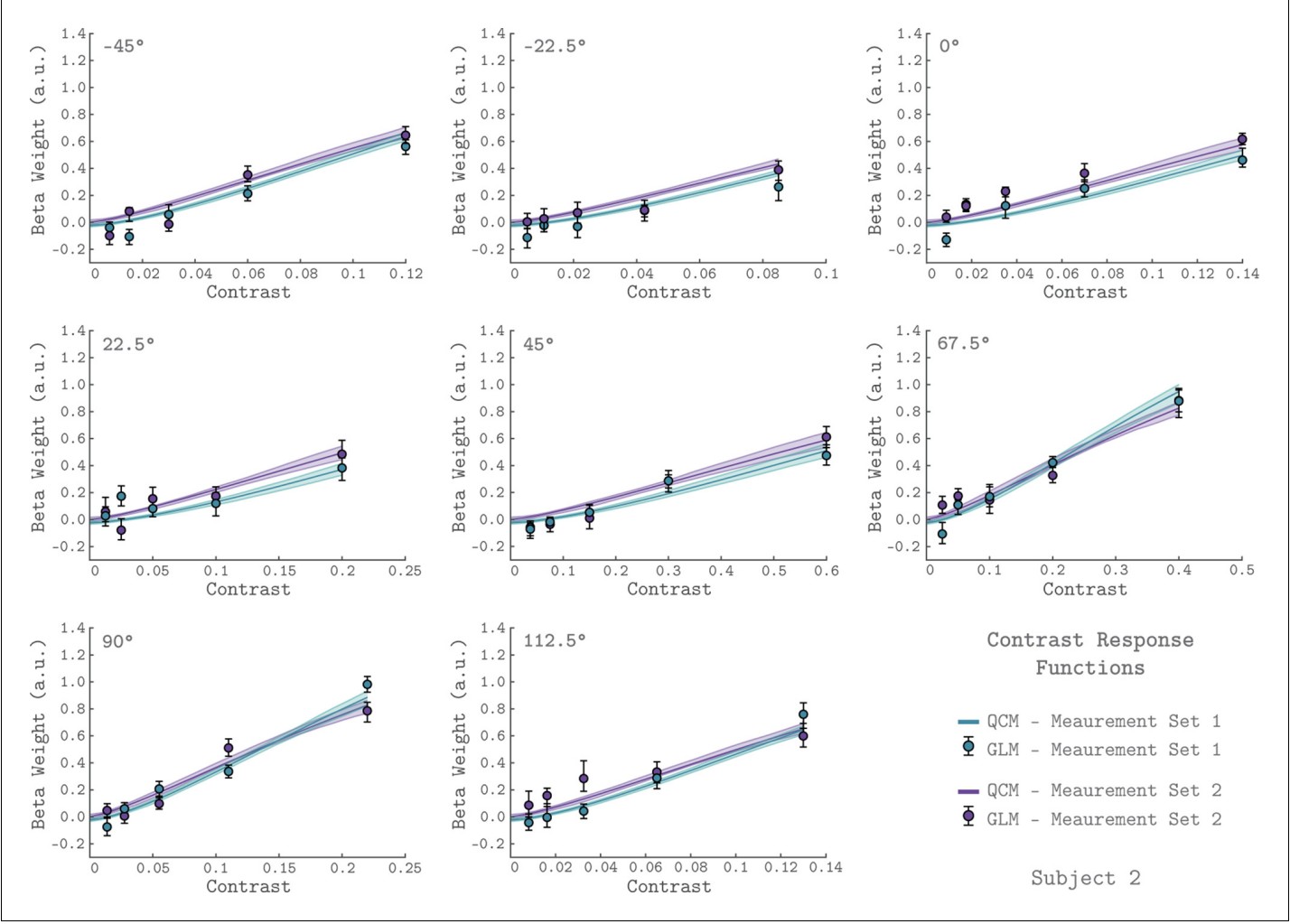

**Figure 2.** V1 contrast response functions for the eight measured chromatic directions from Subject 2. Each panel plots the contrast response function of V1, aggregated over 0° to 20° eccentricity, for a single chromatic direction. The x-axis is contrast, the y-axis is the BOLD response (taken as the GLM beta weight for each stimulus). The chromatic direction of each stimulus is indicated in the upper left of each panel. The curves represent the QCM prediction of the contrast response function. Error bars indicate 68% confidence intervals obtained by bootstrap resampling. Measurement Sets 1 and 2 are shown in green and purple. The x-axis range differs across panels as the maximum contrast used varies with chromatic direction. All data shown have had the baseline estimated from the background condition subtracted such that we obtain a 0 beta weight at 0 contrast.

The online version of this article includes the following figure supplement(s) for figure 2:

**Figure supplement 1.** V1 contrast response functions for the eight measured chromatic directions from Subject 1.

**Figure supplement 2.** V1 contrast response functions for the eight measured chromatic directions from Subject 3.

simplicity, and due to the small differences in fit, we retain the GLM as the point of comparison for the performance of the QCM.

## Quality of GLM time course fit

We examined how well the GLM fit the measured BOLD response from area V1. *Figure 3* shows the fit of the GLM for six example runs from Subject 2. In each panel, the measured BOLD percent signal change is shown as the thin gray line, while the fit obtained from the GLM is shown as the orange line. The orange shaded region represents the 68% confidence interval of the fit found using bootstrap resampling. The GLM fit captured meaningful stimulus-driven variation in the BOLD response, with some variation in fit quality across runs. The median $R^2$ value across runs was 0.41 for Measurement Set 1 and 0.32 for Measurement Set 2. Fits for the other two subjects are provided as *Figure 3—figure supplements 1–2*. Due to the randomized stimulus order within each run, it was not

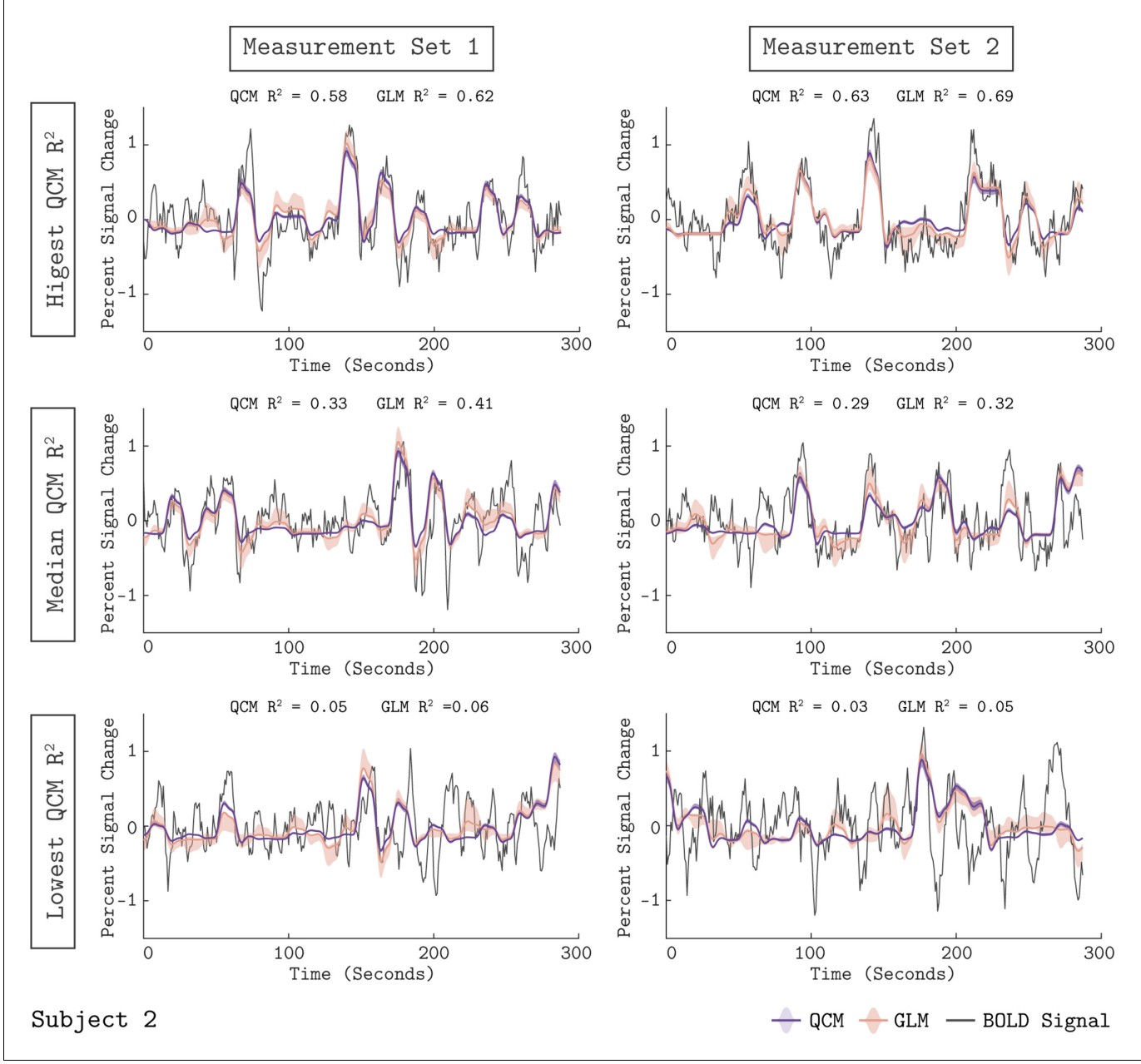

**Figure 3.** Model fits to the V1 BOLD time course. The measured BOLD time course (thin gray line) is shown along with the model fits from the QCM (thick purple line) and GLM (thin orange line) for six runs from Subject 2. Individual runs consisted of only half the total number of chromatic directions. The left column shows data and fits from Measurement Set one and the right column for Measurement Set 2. The three runs presented for each measurement set were chosen to correspond to the highest, median, and lowest QCM $R^2$ values within the respective measurement set; the ranking of the GLM $R^2$ values across runs was similar. The $R^2$ values for the QCM and the GLM are displayed at the top of each panel. The shaded error regions represent the 68% confidence intervals for the GLM obtained using bootstrapping.

The online version of this article includes the following figure supplement(s) for figure 3:

**Figure supplement 1.** Model fits to the V1 BOLD time course from Subject 1.

**Figure supplement 2.** Model fits to the V1 BOLD time course from Subject 3.

**Figure supplement 3.** Mean Residuals for the QCM and the GLM.

straightforward to determine the degree to which the unmodeled variance was due to stimulus-driven structure not modeled by the GLM (e.g. carry-over effects) as opposed to measurement noise. Overall, the quality of the GLM fits supported using the GLM as a benchmark model, as well as using the GLM beta weights as a measure of the V1 response.

## Is the QCM a good model of the BOLD response?

### Characterizing cortical responses with the Quadratic Color Model

The QCM is a parametric special case of the GLM that predicts the BOLD time course using a small number of parameters, and allows for response predictions to modulations in any chromatic direction and contrast in the LM plane. *Figure 2* shows the QCM V1 contrast response functions for Subject 2 (the solid lines). The green and purple lines represent fits to Measurement Set 1 and 2, respectively. The shaded region around both lines represent the 68% confidence intervals for the fits obtained using bootstrap resampling. The QCM contrast response functions agree well with the beta weights obtained from the GLM. The QCM contrast response functions increase monotonically with contrast in all chromatic directions, potentially smoothing measurement variability in the GLM beta weights. There was excellent agreement between the fits to both measurement sets for Subject 2. Similar agreement between the QCM and the GLM and between measurement sets was found for the other two subjects (*Figure 2—figure supplements 1–2*).

We assessed the quality of the QCM fit to the V1 BOLD time course. The purple line in *Figure 3* shows the QCM fit to the BOLD time course with the shaded region representing the 68% confidence interval obtained using bootstrapping. The QCM fit of the time course was of similar quality to the GLM fit. Importantly, the QCM fit was based on only six free parameters, compared to the 41 free parameters of the GLM. Similar quality of fits for QCM can be seen for the other two subjects in *Figure 3—figure supplements 1–2*.

## Comparison of GLM and QCM

We used a leave-runs-out cross-validation procedure to compare the GLM and the QCM (see Materials and methods section for details). This cross-validation compares the ability of the models to predict data not used to fit the parameters, accounting for the possibility that more flexible models (such as the GLM) may overfit the data. *Figure 4* shows the results of the cross-validation comparison for all subjects. Both models track meaningful variation in the signal, although less so for the

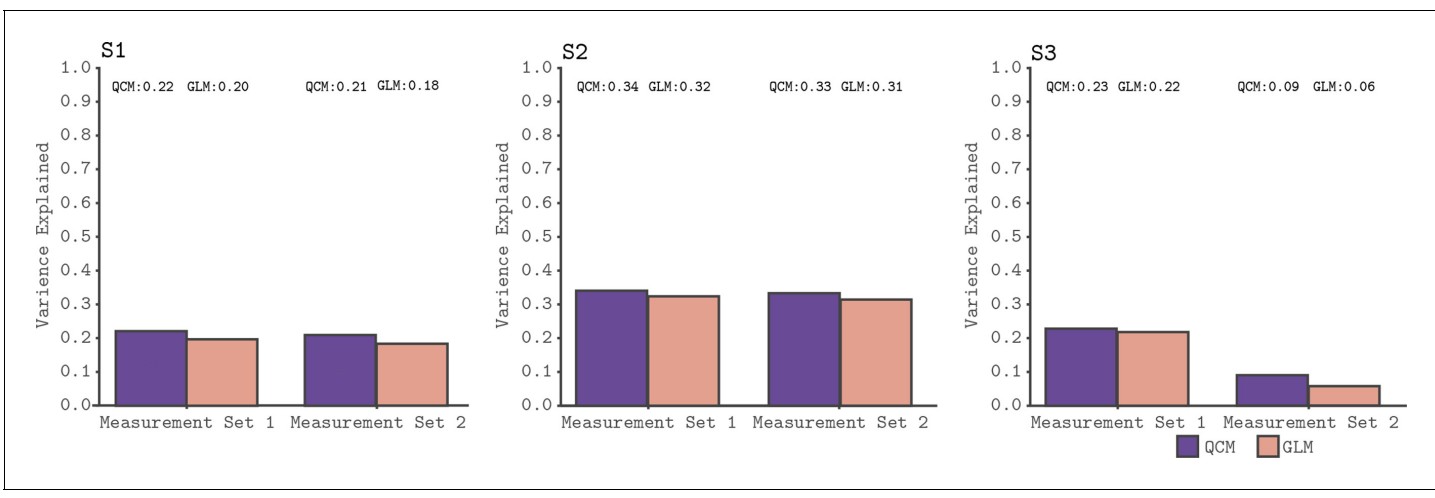

**Figure 4.** Cross-validated model comparison for the QCM and the GLM, from the V1 ROI and for all three subjects. In each panel, the mean leave-one-out cross-validated R$^2$ for the QCM (purple bars) and the GLM (orange bars). These values are displayed at the top of each panel. Within each panel, the left group is for Measurement Set 1 and the the right group is for Measurement Set 2.

The online version of this article includes the following figure supplement(s) for figure 4:

**Figure supplement 1.** Cross-validated model comparison for all models, from the V1 ROI.

data from Subject three in Measurement Set 2. Importantly, we see that the QCM cross-validated $R^2$ is essentially indistinguishable from the GLM cross-validated $R^2$, although in all cases slightly higher.

To further assess differences between the GLM and the QCM, we analyzed the model residuals as a function of the stimulus condition, to check for systemic patterns in the residuals as well as any differences between the two models in this regard. We first examined each direction/contrast pair separately by plotting the residuals of the GLM and QCM over the 14 TRs after the start of each stimulus block. We did not observe any systematic variation in the residuals as a function of contrast level within a single chromatic direction. Therefore, we examined the mean residual value, taken from 4 to 14 TRs after stimulus onset, for all trials in a chromatic direction (collapsed over contrast). We plot these mean residuals for both the GLM and the QCM, as a function of chromatic direction, for each subject and session in *Figure 3—figure supplement 3*. From this, we observe no consistent pattern of residuals within or across models. Note that the residual values for the GLM and QCM mostly overlap, despite the GLM having separate parameters for each stimulus direction.

## QCM generalization

We also employed a leave-session-out cross-validation procedure to assess the generalizability of the QCM (See Materials and methods for details). Given that Sessions 1 and 2 do not share any common chromatic directions, we were able to evaluate how effectively the QCM generalizes to chromatic directions not used to derive the model parameters. The green contrast response functions shown in *Figure 5* result from fitting the QCM to either Session 1 or Session 2, and predicting the responses from the held-out session. The generalization from Session 1 to Session 2 (right-hand subplots) is excellent for this subject. The generalization from Session 2 to Session 1 is also good, albeit with a large confidence interval for the 45° direction. For other subjects and measurement sets, the QCM generalizes reasonably well (*Figure 5—figure supplements 1–3*). Overall, generalizations

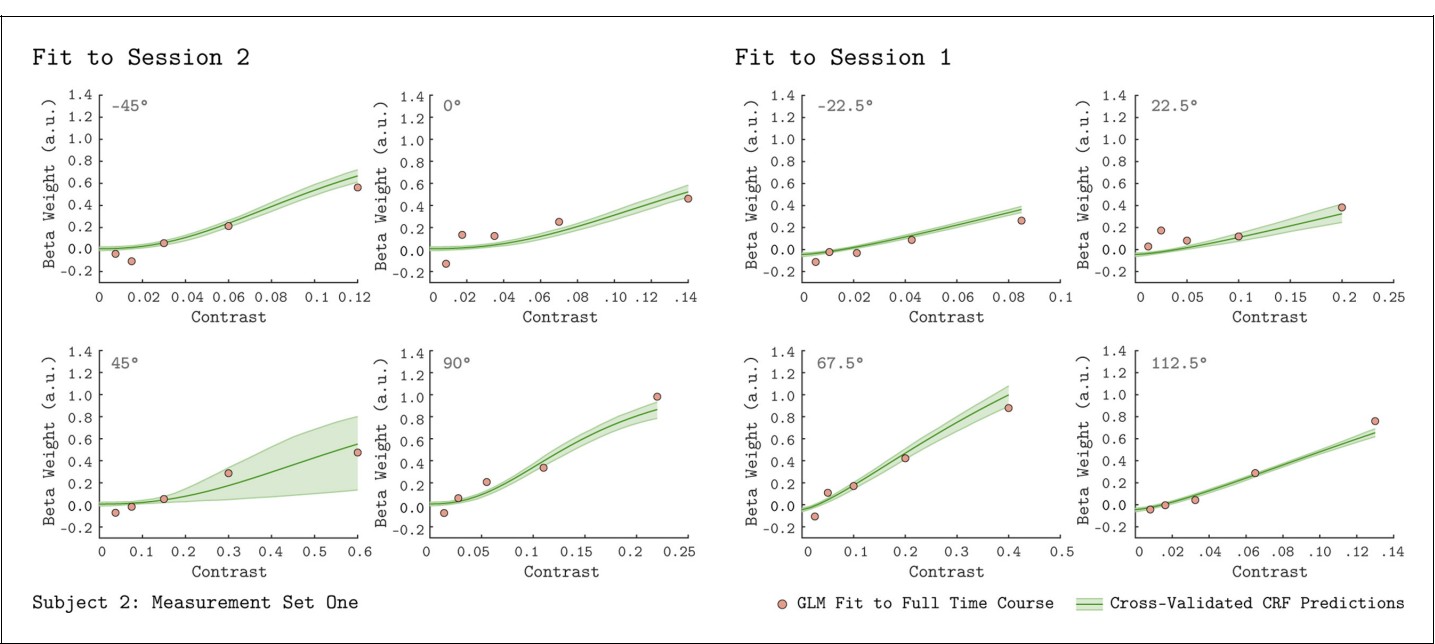

**Figure 5.** Leave-sessions-out cross validation. The contrast response functions in each panel (green lines) are the result of a leave-sessions-out cross-validation to test the generalizability of the QCM. The QCM was fit to data from four out of the eight tested chromatic directions, either from Session 1 or Session 2. The fits were used to predict the CRFs for the held out four directions. The orange points in each panel are the GLM fits to the full data set. The data shown here are for Subject 2, Measurement Set 1. The shaded green error regions represent the 68% confidence intervals for the QCM prediction obtained using bootstrapping. See *Figure 5—figure supplements 1–3* for cross-validation plots from other subjects and measurement sets. The online version of this article includes the following figure supplement(s) for figure 5:

**Figure supplement 1.** Leave-sessions-out cross validation for Subject 1.
**Figure supplement 2.** Leave-sessions-out cross validation for Subject 2.
**Figure supplement 3.** Leave-sessions-out cross validation for Subject 3.

from Session 1 to Session 2 perform better than those from Session 2 to Session 1. This finding may reflect the particular set of chromatic directions presented in each session: only Session 1 includes a chromatic direction close to the major axis of the ellipse, which better constrains the QCM fit. Therefore, the QCM is capable of generalizing well to unmeasured chromatic directions, with the requirement that the stimuli include chromatic directions and contrasts that adequately constrain the model parameters.

## QCM characterization of V1 BOLD response

Conceptually, the parameters of the QCM characterize two key model components. The first component defines the contrast-independent shape of elliptical isoresponse contour. This describes the relative sensitivity of V1 to modulations in all chromatic directions within the LM contrast plane. The second component defines the response nonlinearity, which is independent of chromatic direction. It operates on equivalent contrast to produce the underlying neural response.

### Elliptical isoresponse contours

The isoresponse contour is described by two parameters: the direction of least sensitivity (ellipse angle; counterclockwise to the positive abscissa) and the ratio of vector lengths between the most and least sensitive directions (minor axis ratio; see Quadratic Color Model Section and Appendix 1). Within the QCM, the angle and minor axis ratio provide a complete description of chromatic sensitivity that is contrast independent.

*Figure 6* shows the QCM isoresponse contours for all three subjects and both measurement sets. We found that for all subjects and measurement sets, the angle of the isoresponse contours was oriented at approximately 45°. An ellipse angle of 45° indicates that V1 was least sensitive to stimuli modulated in the L+M direction, and most sensitive to stimuli modulated in the L-M direction. Across all subjects and measurement sets, the minor axis ratio parameters ranged between 0.15 and 0.25. Thus, for the spatial and temporal properties of our modulations, V1 was roughly five times more sensitive to modulations in the L-M direction than the L+M direction. We found good agreement between the isoresponse contours from the independent measurement sets as well as across subjects.

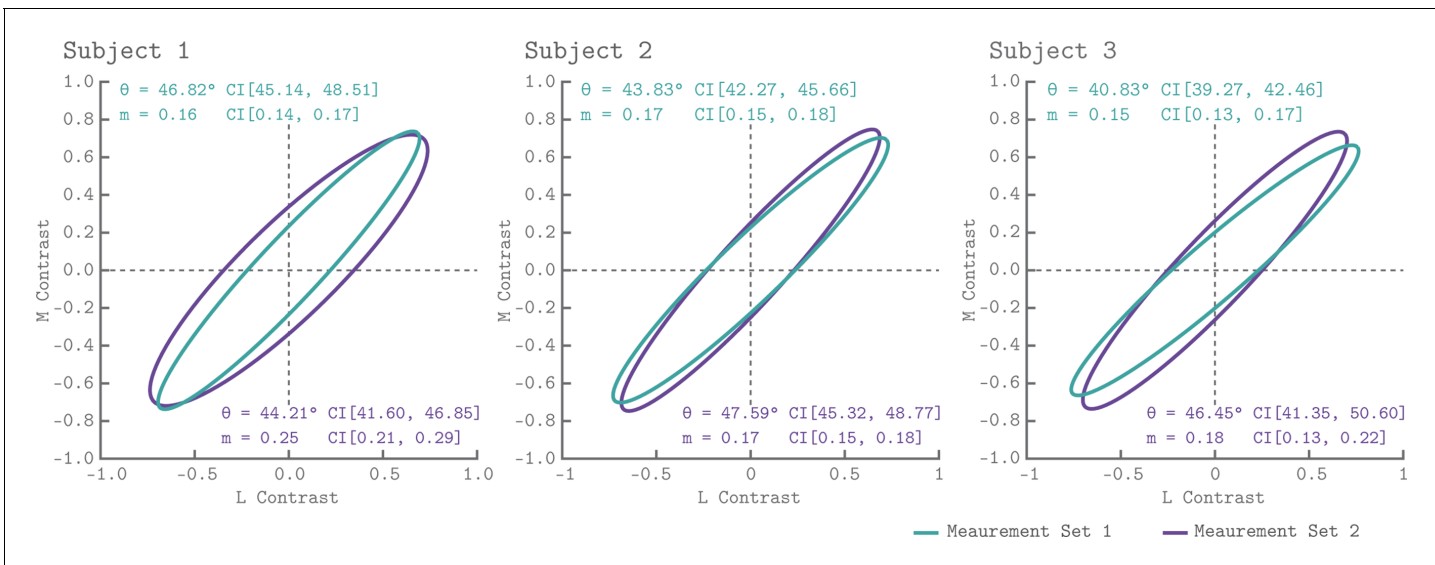

**Figure 6.** V1 isoresponse contours. The normalized elliptical isoresponse contours from the QCM are plotted, for each subject, in the LM contrast plane. The green ellipses show the QCM fits to Measurement Set 1 and the purple ellipses show fits to measurement 2. The angles and minor axis ratios along with their corresponding 68% confidence intervals obtained using bootstrapping are provided in the upper left (Measurement Set 1) and lower right (Measurement Set 2) of each panel.

The online version of this article includes the following figure supplement(s) for figure 6:

**Figure supplement 1.** Isoresponse Contours for the LCM and the QCM.

## Equivalent contrast nonlinearity

*Figure 7* shows the V1 equivalent contrast nonlinearity of the QCM for Subject 2 for both measurement sets. This non-linearity describes how the underlying neuronal response increases with increasing equivalent contrast. We used the isoresponse contour of the QCM to convert the chromatic direction and cone contrast of each stimulus to its equivalent contrast. This allowed us to replot each beta weight derived from the GLM (*Figure 2*) on an equivalent contrast axis (*Figure 7*; closed circles). For all subjects, the single nonlinearity accurately captured the dependence of the GLM beta weight on equivalent contrast, with no apparent bias across chromatic directions. The agreement between the GLM beta weight points and QCM fits demonstrated that separating the effects of chromatic direction and contrast in the QCM is reasonable. *Figure 7—figure supplement 1* provides the same plots for Subjects 1 and 3.

## Dependence of chromatic sensitivity on eccentricity

### Isoresponse contour parameter maps

There is considerable interest in how sensitivity to modulations in the LM contrast plane varies with eccentricity. Understanding such variation is important both for describing visual performance and for drawing inferences regarding the neural circuitry that mediates color vision. Since the QCM separates chromatic sensitivity from the dependence of the response on contrast, examining how the shape of the QCM isoresponse contour varies with eccentricity addresses this question in a contrast-independent manner. We fit the QCM to the BOLD time course of each vertex in the template map of visual areas developed by *Benson et al., 2014*. This allowed us to visualize how the parameters that describe the isoresponse contour varied with eccentricity within V1.

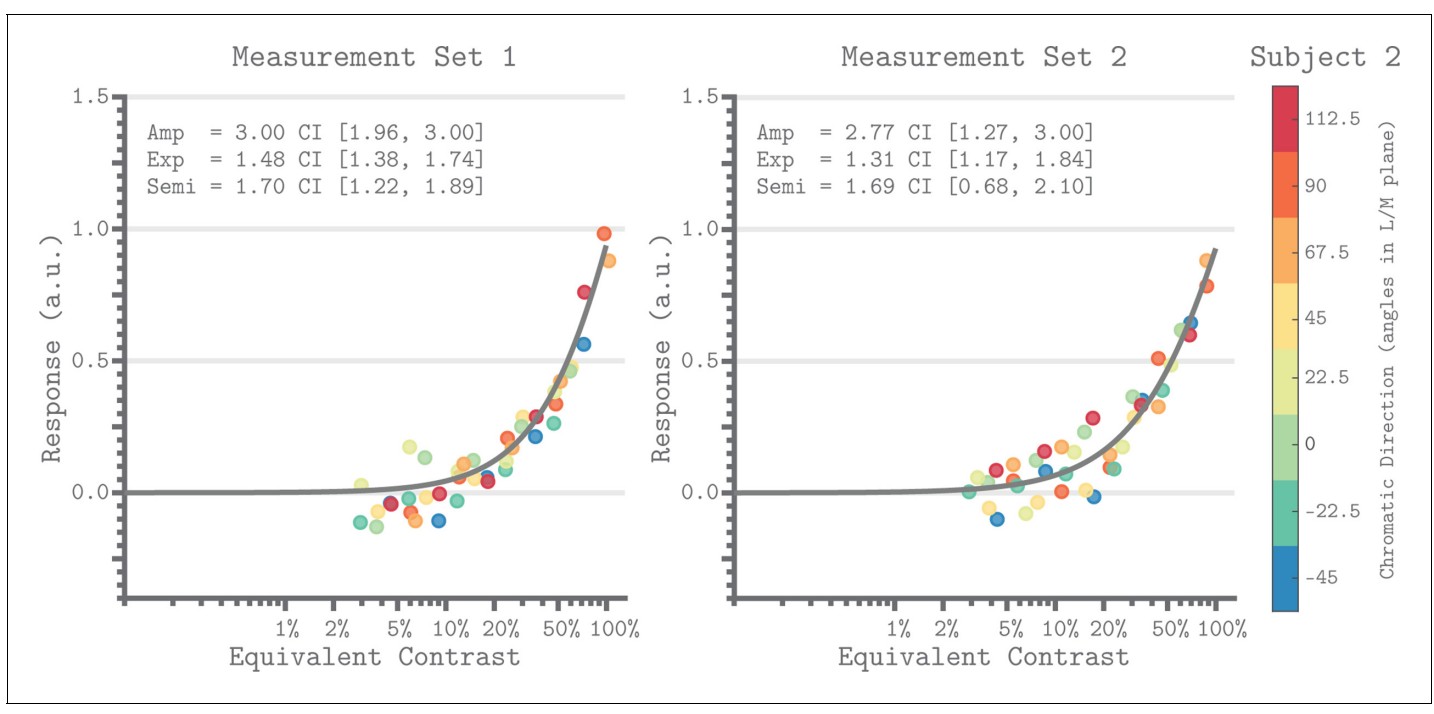

**Figure 7.** Equivalent contrast non-linearities of the QCM for V1 from Subject 2. The x-axis of each panel marks the equivalent contrast and the y-axis is the neuronal response. The gray curve in each panel is the Naka-Rushton function obtained using the QCM fit. These curves show the relationship between equivalent contrast and response. The parameters of the Naka-Rushton function are reported in upper left of each panel along with the 68% confidence intervals obtained using bootstrapping. The points in each panel are the GLM beta weights mapped via the QCM isoresponse contours of Subject 2 onto the equivalent contrast axis (see Appendix 1). The color of each point denotes the chromatic direction of the stimuli, as shown in the color bar. The left panel is for Measurement Set 1 and the right panel is for Measurement Set 2. Note that our maximum contrast stimuli do not produce a saturated response. Note that our stimuli did not drive the response into the saturated regime.

The online version of this article includes the following figure supplement(s) for figure 7:

**Figure supplement 1.** Equivalent Contrast Non-Linearities of the QCM for V1.

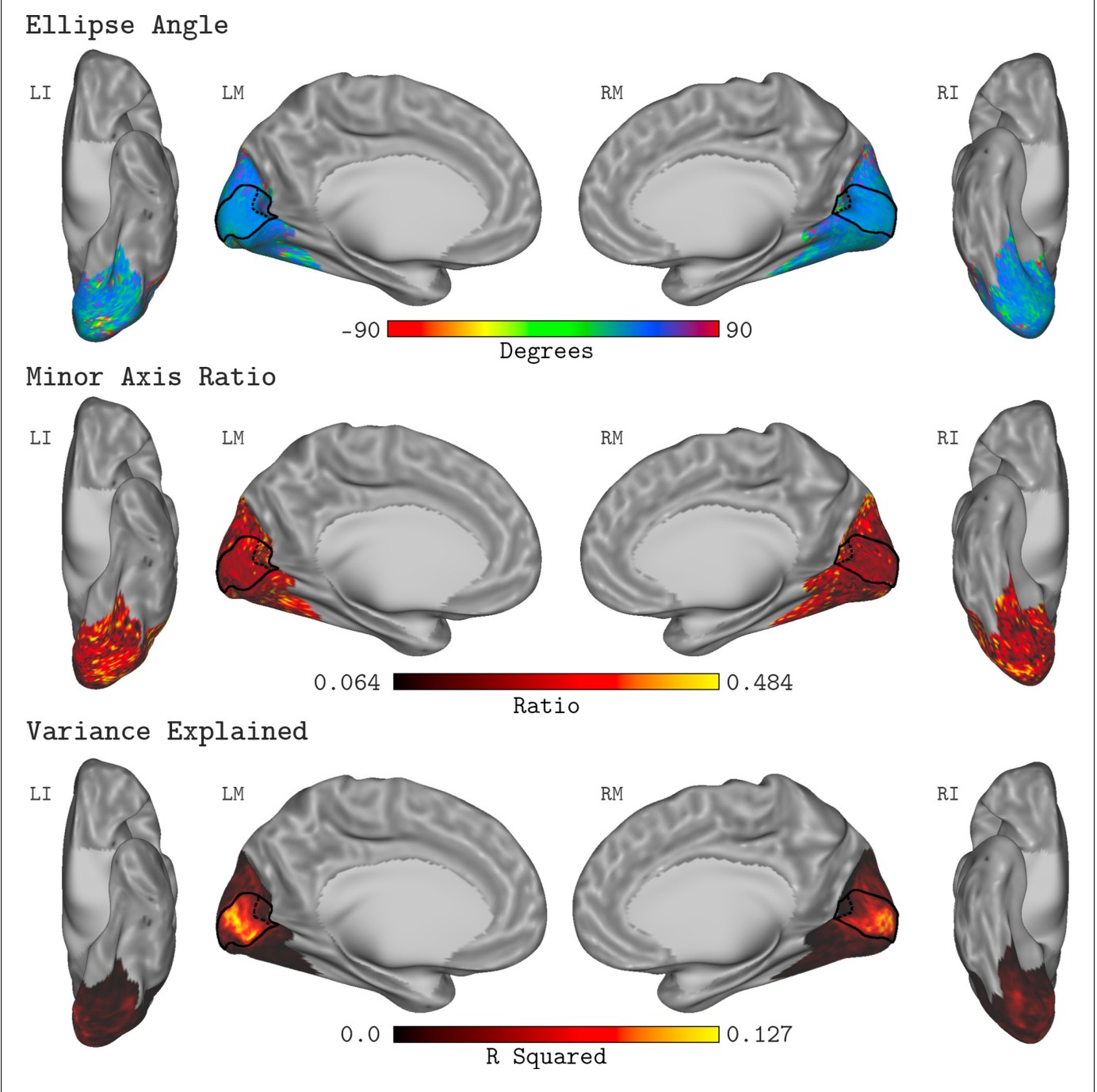

**Figure 8.** QCM average parameter maps. The QCM parameters, fit at all vertices within the visual cortex mask, averaged across all subjects and measurement sets. The top, middle, and bottom rows show maps of the average ellipse angle, minor axis ratio, and variance explained, respectively. The scale of the corresponding color map is presented below each row. The nomenclature in upper left of each surface view indicates the hemisphere (L: left or R: right) and the view (I: inferior, L: lateral, or M: medial). The medial views show the full extent of the V1 ROI on the cortical surface (denoted by the solid black outline). The 20° eccentricity boundary used to define the V1 ROI used for all analyses is shown by the black dashed line. The online version of this article includes the following figure supplement(s) for figure 8:

**Figure supplement 1.** Average $R^2$ map for the GLM for early visual cortex.

*Figure 8* shows the QCM parameter maps for the ellipse angle, the minor axis ratio, and the variance explained displayed on the cortical surface. Here, the data were averaged across all subjects

and measurement sets. In all panels, the full extent of V1 is denoted by the black outline on the cortical surface, while the 20° eccentricity ROI used in the V1 analyses above is shown by the black dashed line. Apparent in the maps is that neither parameter varied systematically within V1, a feature of the data that is consistent across measurement sets and subjects. Outside of V1, the $R^2$ values were markedly lower, and there was higher variability in the QCM parameters.

We further examined the variance explained by the GLM, fit to every vertex on the cortical surface. Within early visual cortex (EVC, the spatial extent of the Benson template), we did not observe differences in $R^2$ larger than 0.03 in non-cross-validated model fits between the GLM and the QCM (GLM – QCM). We generally found that the variance explained by the GLM in vertices outside of V1 was close to zero, with the exception of a small patch of values in the vicinity of hV4/VO1. The GLM variance explained in this area was roughly half of that explained within V1. To more fully characterize these regions, we fit the QCM to the median time course from the subject specific registrations of hV4 and VO1 as defined by the retinotopic atlas from *Wang et al., 2015* (implemented in Neuropythy). The parameters of the QCM fit for hV4 and VO1 were generally consistent with those found for V1, although fit quality was worse. Overall, as our spatially uniform stimuli were not highly effective at eliciting reliable responses outside of V1, we refrain from drawing definitive conclusions about responses outside of V1. The average variance explained map within EVC for the GLM is shown in *Figure 8—figure supplement 1*.

## No change in V1 chromatic sensitivity with eccentricity

We leveraged the QCM to examine how chromatic sensitivity varies with eccentricity within V1. *Figure 9* plots the V1 QCM parameters as a function of eccentricity, for Subject 2. The left panel shows the minor axis ratio and the right panel shows the ellipse angle. In both plots, individual points represent a single vertex, with the x-axis giving the visual field eccentricity of that vertex obtained from the *Benson et al., 2014* template, and the y-axis giving the parameter value. The transparency of each point indicates the $R^2$ value of the QCM fit for the corresponding vertex. The maximum $R^2$ value across vertices for Measurement Sets 1 and 2 were 0.25 and 0.24, respectively. The lines in each panel reflect a robust regression fit to the points. We found that there is little change in either

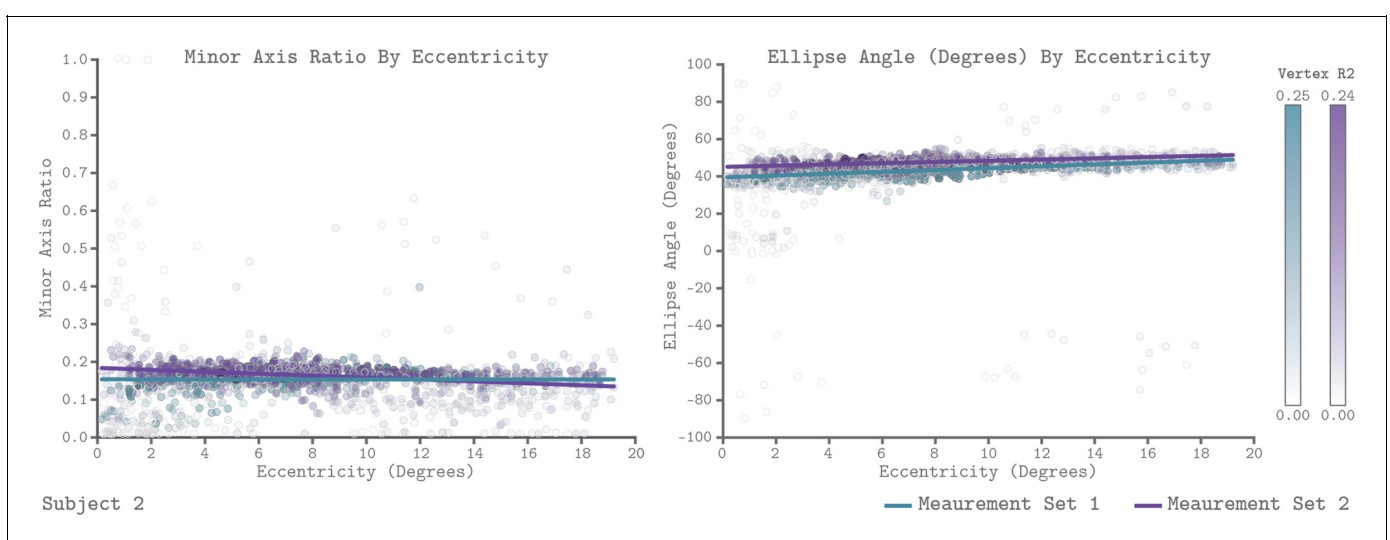

**Figure 9.** QCM parameters as a function of eccentricity for Subject 2. The left and right panels show scatter plots of the minor axis ratio and ellipse angle plotted against their visual field eccentricity, respectively. Each point in the scatter plot shows a parameter value and corresponding eccentricity from an individual vertex. Green indicates Measurement Set 1 and purple indicates Measurement Set 2. The lines in each panel are robust regression obtained for each measurement set separately. The transparency of each point provides the $R^2$ value of the QCM at that vertex. The color bars provide the $R^2$ scale for each measurement set.

The online version of this article includes the following figure supplement(s) for figure 9:

**Figure supplement 1.** L+M and L-M responses predicted using QCM as a function of eccentricity.

**Figure supplement 2.** QCM parameters as a function of eccentricity.

parameter with eccentricity. For Subject 2, the best fit lines had slightly negative slopes for the minor axis ratio and slightly positive slopes for the ellipse angle, with good agreement across measurement sets. The overall change in parameter values from 0° to 20°, however, was small compared to the vertical spread of values at each eccentricity. We compared the change in parameter values from 0° to 20° to the variability across measurement sets for all three subjects (*Table 1* for minor axis ratio, and *Table 2* for ellipse angle). Across subjects, the majority of sessions showed small differences in parameter values from 0° to 20°, but we note that these did in some cases exceed the measurement set-to-set difference in the parameter values obtained for all of V1.

The plots shown in *Figure 9* examine how the QCM parameters vary with eccentricity. To allow comparison with prior studies of how the BOLD response varies with eccentricity within V1 (*Vanni et al., 2006*; *Mullen et al., 2007*; *D'Souza et al., 2016*), we also used the QCM to predict how the response would vary for stimuli in the L+M and the L-M directions, and plot these predicted responses as a function of eccentricity (*Figure 9—figure supplement 1*). This was done on a vertex-by-vertex basis, using within-vertex QCM parameters. Specifically, we chose the 50% contrast stimulus condition for both the L-M and the L+M direction (contrasts of 0.06 and 0.30 respectively). Using these stimuli, we computed the predicted neuronal response by applying the QCM forward model (the transformation to equivalent contrast and the Naka-Rushton steps) using the parameter values corresponding to that particular vertex. Examining the data in this way reveals a negligible change in response as a function of eccentricity for both the L+M and L-M directions, for all subjects and measurement sets. We return in the discussion to consider the relation between our results and those found in prior studies.

## Discussion

We develop a quantitative model of the visual cortex response to chromatic stimuli in the LM contrast plane, the quadratic color model (QCM), and examine its ability to fit V1 BOLD fMRI responses to spatially uniform chromatic stimuli. We find that the QCM accounts for the same cross-validated variance as a conventional GLM, with far fewer free parameters (6 as compared to 41). The model generalizes across both chromatic direction and contrast to predict V1 responses to a set of stimuli that were not used to fit the model parameters. The experiment was replicated for each subject using the same stimuli across separate measurement sets. Both the data and the model fits replicate well for each subject and are similar across subjects, giving us confidence in the power of the measurements.

The QCM is a separable model with respect to the effects of chromatic direction and contrast. This allowed us to evaluate the chromatic sensitivity in V1 of our subjects in a manner that is independent of the effects of contrast. We find that V1 is most sensitive to L-M contrast modulations and least sensitive to L+M contrast modulations, when contrast is defined using vector length in the LM contrast plane. This was shown in all subjects and measurement sets by the isoresponse contours

**Table 1.** Robust regression line parameters summarizing the change in minor axis ratio with eccentricity for all subjects.

These parameters are the same as seen for Subject 2 in *Figure 9*. The subject and set columns indicate the subject and measurement set of the robust regression fit. The slope and offset column show the parameters of the regression line. The Δ 0° to 20° column is the magnitude of the change in minor axis ratio between 0° and 20° eccentricity. The Δ Set to Set column shows the absolute difference in the minor axis ratio fit to the V1 median time course between Measurement Set 1 and 2.

| Subject | Set | Slope | Offset | Δ 0° to 20° | Δ Set to Set |
|---------|-----|-------|--------|-------------|--------------|
| S1 | 1 | −1.19e-3 | 0.163 | 0.0238 | 0.09 |
| S1 | 2 | −4.17e-4 | 0.24 | 0.0084 | |
| S2 | 1 | −7.54e-6 | 0.154 | 0.0002 | 0.00 |
| S2 | 2 | −2.51e-3 | 0.183 | 0.0504 | |
| S3 | 1 | −3.27e-3 | 0.114 | 0.0654 | 0.03 |
| S3 | 2 | −3.9e-4 | 0.158 | 0.0078 | |

**Table 2.** Robust regression line parameters summarizing the change in ellipse angle with eccentricity for all subjects.
Columns are formatted the same as *Table 1*.

| Subject | Set | Slope | Offset | Δ 0° to 20° | Δ Set to Set |
|---------|-----|-------|--------|-------------|--------------|
| S1 | 1 | 0.039 | 46.2 | 0.78 | 2.61 |
| S1 | 2 | 0.313 | 41.4 | 6.26 | |
| S2 | 1 | 0.496 | 39.5 | 9.92 | 3.76 |
| S2 | 2 | 0.330 | 45.1 | 6.60 | |
| S3 | 1 | 0.247 | 39.3 | 4.94 | 5.62 |
| S3 | 2 | 0.425 | 43.3 | 8.50 | |

of each subject being oriented at approximately 45° and having a minor axis that is roughly five times smaller than the major axis. This result is broadly consistent with previous fMRI studies of V1 chromatic contrast sensitivity, although the exact sensitivity ratio varies with the spatial and temporal properties of the stimuli (*Engel et al., 1997*; *Liu and Wandell, 2005*; *Mullen et al., 2007*; *D'Souza et al., 2016*; *Mullen et al., 2010a*). By considering cortical responses in terms of the parameters of the QCM fit, we are able to provide a quantitative account of chromatic sensitivity, as opposed to a categorical assignment of voxels as 'color' or 'luminance' responsive.

The QCM also allows us to examine the equivalent contrast response nonlinearity, although doing so is not the focus of this paper. This non-linearity depends on chromatic direction only through a direction-dependent contrast gain that is captured by the isoresponse contour. This can be observed in *Figure 7* through the overlap of the non-linearity and the transformed GLM beta weights. Although we used the Naka-Rushton function to fit the nonlinearity, this was a choice of convenience and the precise shape of the non-linearity is not strongly constrained by our data set. This is because our stimuli did not drive the response into the saturating regime (*Figure 7*, *Figure 7—figure supplement 1*). A stronger test of the contrast/direction separability embodied by the QCM, as well as stronger constraints on the shape of the non-linearity, would be provided by stimuli that drive the V1 response to saturation.

Neither the QCM nor the GLM explain all of the variance in the data. Since our experimental design did not involve multiple measurements with the same stimulus sequence (stimulus sequences were randomized across runs and measurement sets), we cannot untangle the degree to which the unexplained variance is due to systematic but unmodeled aspects of the response or to measurement noise. In comparing our reported $R^2$ values to those in other studies, it is important to bear in mind that $R^2$ values are expected to be higher in cases where the signal being fit is the average time course over multiple runs with the same stimulus sequence, as compared to when the $R^2$ values are computed with respect to fits to individual runs, even if there is no difference in the quality of the underlying response model. Another factor that can affect $R^2$ is un-modeled physiological effects on the BOLD signal due to blinking, breathing, heart beats, etc. We did not collect eye tracking measurements or pulse oximetry, so we cannot model such effects.

In certain cases, attentional task difficulty can modulate BOLD responses (*Kay and Yeatman, 2017*). We employed only one level of attentional task difficulty and thus do not have data on how varying the attentional task might affect the responses we measured. We do not, however, have any particular reason to think that the chromatic tuning and contrast response functions we measured would have been substantially different in the context of different attentional task difficulty. In this regard, we note that *Tregillus et al., 2021* measured contrast response functions for L-M and S-(L+M) color directions within V1 under two different attentional tasks and found no significant effect of task on the two contrast response functions.

## Relation to psychophysics

A goal of systems neuroscience is to link measurements of neuronal properties to measurements of behavior. To make these links, the measurements made in each domain must be placed into a common space for comparison. The QCM provides a way to represent fMRI measurements in a manner

that makes such comparisons straightforward. The contrast-invariant isoresponse contour from the QCM provides us with a stimulus-referred characterization of the BOLD fMRI response. Other methodologies, such as psychophysics or electrophysiology, may be used to obtain similar characterizations, allowing for comparisons across response measures within this common framework. For example, an approach to studying chromatic sensitivity is to characterize the isothreshold contour, which specifies the set of stimulus modulations that are equally detectable. *Engel et al., 1997* took this approach and found that for low temporal frequencies the psychophysical isothreshold and BOLD fMRI isoresponse contours in the LM contrast plane were well-described as ellipses and had similar shapes. While some work has argued that psychophysical isothreshold may deviate subtly from ellipses (for review see *Stockman and Brainard, 2010*), two studies that attempted to reject the elliptical form of such contours did not do so (*Poirson et al., 1990*; *Knoblauch and Maloney, 1996*). Consistent with Engel, Zhang, and Wandell (1997), we found elliptical BOLD isoresponse contours at our 12 Hz temporal frequency with highest sensitivity in the L-M direction. As they note, although psychophysical isothreshold contours remain well-described by ellipses at higher temporal frequencies, their orientation changes to favor L+M sensitivity over L-M sensitivity. This dissociation in the particulars of the isothreshold and BOLD isoresponse contours makes it unlikely that the mechanisms that contribute to the BOLD response in V1 limit psychophysical detection at the higher temporal frequencies, unless there are important temporal-frequency dependent changes in response variability that are not captured by the BOLD measurements.

## Relation to underlying mechanisms

Many theories of color vision postulate that signals from the L-, M-, and S-cone photoreceptors are combined to form three post-receptoral mechanisms, roughly characterized as an additive combination of L- and M-cone contrast (L+M), an opponent combination of L- and M-cone contrast (L-M), and an opponent combination of S-cone contrast with L- and M-cone contrasts (S-(L+M)) (*Stockman and Brainard, 2010*; *Shevell and Martin, 2017*). Our finding that the major and minor axes of the isoresponse ellipse are well-aligned with the L+M and L-M modulation directions agrees with such theories. More generally, a quadratic isorepsonse contour can be produced by a quadratic mechanism that computes a sum of the squared responses of two underlying linear mechanisms, where the output of each linear mechanism is a weighted sum of L- and M-cone contrasts (*Poirson et al., 1990*). If the two linear mechanisms are L+M and L-M mechanisms with the weights appropriately chosen to represent the relative sensitivities (L-M sensitivity greater than L+M sensitivity), then the isoresponse contour of the resulting quadratic mechanism will be a close match to those we measured.

Note, however, that other pairs of underlying mechanisms are also consistent with the same elliptical isoresponse contours (*Poirson et al., 1990*), so that our isoresponse contours do not uniquely determine the sensitivity of the underlying linear mechanisms, even within the QCM together with the assumption that there are two such mechanisms.

More generally, one can construct non-quadratic models whose isoresponse contours approximate the ellipse we found using the QCM, and if this approximation is good our data will not reject such models. To illustrate this point, we developed and fit an alternate model, the Linear Channels Model (LCM), a variation on the Brouwer and Heeger channel model (*Brouwer and Heeger, 2009*; *Kim et al., 2020*), that accounts for our data about as well as the QCM (see Appendix 1; *Figure 4—figure supplement 1*). The best fitting isoresponse contours found with the LCM, which could in principle deviate considerably from an ellipse, none-the-less approximate the ellipse we found using the QCM, but are not perfectly elliptical (*Figure 6—figure supplement 1*). Despite the agreement at the functional level of the isoresponse contours, the properties of the mechanisms underlying the LCM differ from those of the QCM, and these properties also differ across different instantiations of the LCM that account for the data equally well (see Appendix 1).

Because of the similarity in cross-validated $R^2$ values across the GLM, Naka-Rushton, QCM and LCM models, the reader may wonder whether the data have sufficient power to reject any isoresponse contour shape. To address this, we fit and cross-validated a form of the QCM with the angle constrained to 0 degrees. This resulted in a noticeably lower cross-validated $R^2$ for this model as compared to all other models we tested (*Figure 4—figure supplement 1*, labeled at 'QCM locked') and provides reassurance that the data indeed have power to inform as to the shape of the

isoresponse contour. We expect that other isoresponse contour shapes that differ from the best fitting QCM contour to a degree similar to that of the constrained ellipse would also be rejected.

Thus, while measurements of the BOLD response place constraints on the population response properties of the neuronal mechanisms, these properties are not uniquely determined given the BOLD response alone. The ambiguity is further increased if we consider properties of individual neurons, as the aggregate BOLD response will be shaped both by the response properties of such neurons and the numbers of different types of neurons in the overall neural population. With that caveat, we make some general observations. Our stimuli were large, spatially uniform (effectively 0 c.p.d.) chromatic modulations that were temporally modulated at 12 Hz. These stimuli could drive 'color', 'luminance' and 'color-luminance' cells as described by *Johnson et al., 2001*, depending on the particular chromatic direction. The 0 c.p.d. stimuli would presumably produce strong responses in the 'color' cells for our L-M direction, given that these cells are thought to behave as low-pass filters in the spatial domain and have unoriented receptive fields. It is less clear how strongly 'luminance' and 'color-luminance' cells would respond to our spatially uniform stimuli given that these cells are spatially bandpass. *Schluppeck and Engel, 2002* plot the spatial frequency response function estimated from the data from *Johnson et al., 2001*. These functions plot the average firing rate as a function of spatial frequency for the color, color-luminance, and luminance cells. Taking the lowest spatial frequency present in the dataset (0.1 c.p.d.), firing rates for color cells are roughly 5x times higher than the firing rates for luminance and color-luminance cells. This is the same as our average minor axis ratio which indicates V1 is roughly five times more sensitive to L-M than to L+M. A caveat here is that the 12 Hz flicker rate of our stimuli might shift responses relative to the analysis of *Schluppeck and Engel, 2002*.

It is also important to note that our data do not distinguish the extent to which the response properties of the BOLD signals we measure in V1 are inherited from the LGN or are shaped by processing within V1. The spatiotemporal properties of our stimuli would robustly drive cells in cell in the LGN that project to V1 (*Lankheet et al., 1998*). Even though the signals measured in our experiment are spatially localized to V1, we cannot ascribe the observed response properties to particular neural processing sites. As such, the V1 sensitivities found from fitting the QCM may be inherited from areas prior to V1. In principle, they could also be affected by feedback from other cortical areas. We also undertook an analysis of data from the LGN, but found that these signals were too noisy to reveal reliable stimulus-driven responses in our data.

Finally, it is interesting to observe that quadratic models have been used to characterize the isoresponse properties of individual neurons in macaque V1 (*Horwitz and Hass, 2012*). Roughly half of the neurons tested in this paper were best fit by quadratic isoresponse surfaces (in the L-, M-, and S-cone contrast space) while the other half were well fit by a linear model whose isoresponse surfaces were parallel planes. Of the quadratic isoresponse surfaces, some were ellipsoidal while others were hyperbolic. As noted above, despite this qualitative similarity, connecting the diverse population of individual neural responses to the aggregated BOLD fMRI response remains a challenge for future work. The QCM fit to our data aids in this endeavor through the constraint its isoresponse contour places on the aggregated neural response.

## Change of chromatic sensitivity with retinal eccentricity

Many aspects of visual function change with eccentricity (*Rosenholtz, 2016*), and understanding and quantifying this variation is a key part of a functional characterization of vision. In addition, prior work attempts to relate such functional variation with eccentricity to variation in the underlying neural mechanisms. Relevant to the present work is the idea that variation of chromatic sensitivity with eccentricity can inform as to how signals from separate cone classes are combined by retinal and cortical neural circuitry (*Lennie et al., 1991*; *Mullen and Kingdom, 1996*; *Wool et al., 2018*; *Baseler and Sutter, 1997*). In this context, we examined how the parameters of the QCM for individual vertices varied with eccentricity across V1. Overall, we find little change in the isoresponse contours with eccentricity within the central 20° of the visual field (*Figure 9*; *Tables 1* and *2*; *Figure 9—figure supplement 2*), and an analysis of how predicted response to L+M and L-M modulations would change with eccentricity also shows little or no effect (*Figure 9—figure supplement 1*). In addition, we do not observe any clear change in the parameters of the contrast-response function with eccentricity (analysis not shown). Overall, the BOLD response to chromatic modulations, as evaluated in our data using the QCM, is remarkably stable across V1. That the orientation of the elliptical

isoresponse contours does not change with eccentricity is consistent with psychophysical studies that show that the relative L- and M-cone inputs to an L-M mechanism are stable across the visual field (*Newton and Eskew, 2003*; *Sakurai and Mullen, 2006*). The fact that our data do not show a loss in L-M sensitivity relative L+M sensitivity with increasing eccentricity, on the other hand, is not commensurate with psychophysical studies that do show such a loss (*Stromeyer et al., 1992*; *Mullen and Kingdom, 2002*; *Mullen et al., 2005*; *Hansen et al., 2009*). As discussed above, BOLD fMRI sensitivity in V1 does not always mirror psychophysical sensitivity. Thus we focus below on comparison between our results and other fMRI studies of how sensitivity in V1 varies with eccentricity.

Several prior studies have used BOLD fMRI to examine how visual cortex responses vary with eccentricity to stimuli modulated in L-M and L+M directions (*Vanni et al., 2006*; *Mullen et al., 2007*; *D'Souza et al., 2016*). Both *Mullen et al., 2007* and *Vanni et al., 2006* report a decrease in the V1 response to L-M modulations with eccentricity, while the response to L+M remains roughly constant. This differs from our result, and the size of the effects in these papers are large enough that we would expect that if they were present in our data they would be visible in the analysis shown in *Figure 9—figure supplement 1* (see Figure 8 in *Mullen et al., 2007* and Figure 8 in *Vanni et al., 2006*). Both speculate that the mechanism underlying this observation is non-selective (random) connections between L and M cones and retinal ganglion cell receptive fields. If these connections are non-selective, L-M sensitivity would be expected to decrease with eccentricity. This is because the area in which receptive fields pool cone inputs increases with distance from the fovea, progressively reducing the likelihood that random L- and M-cone inputs to the center and surround will produce chromatic opponency (*Lennie et al., 1991*; *Mullen and Kingdom, 1996*; *Wool et al., 2018*). In contrast, *D'Souza et al., 2016* find, for the majority of spatial frequencies studied, no change in L-M response relative to the response to isochromatic luminance modulations. This result is generally in line with our data, although at their lowest spatial frequencies D'Souza and colleagues observe a modest decline in relative L-M sensitivity. Following the same line of reasoning as *Mullen et al., 2007* and *Vanni et al., 2006* but reaching the opposite conclusion, *D'Souza et al., 2016* take their result as supporting the idea that connections between cones and some classes of ganglion cells are selective for cone type and preserve chromatic sensitivity across the retina.

Comparison across our and the prior studies is complicated by variation in the stimuli used. Indeed, a dependency on spatial frequency is indicated by the data of *D'Souza et al., 2016*. We used spatially uniform fields, while other studies use stimuli with higher-spatial frequency content. More generally, other factors could also lead to variation across studies, as well as complicate inferring the properties of retinal wiring from how psychophysical thresholds or measurements of cortical response vary with eccentricity.

One such factor is the changes in cone spectral sensitivity with eccentricity, caused primarily by variation in macular pigment and photopigment optical density. Variation in macular pigment and photopigment optical density can produce eccentricity-dependent deviations in the degree of actual cone contrast reaching the photoreceptors. Prior studies do not account for this variation, leading to the possibility that effects of eccentricity on sensitivity are due to receptoral, rather than post-receptoral mechanisms. In our study, we designed spectral modulations that produce the same contrasts in both 2-degree and 15-degree cone fundamentals (see Materials and methods and *Tables 5–7*). This reduces the change in cone contrast with eccentricity for our stimuli.

Another factor, not emphasized in previous work, is that the size of the effects will depend on where on the underlying contrast response functions the responses to the stimuli in the chromatic directions being compared lie. To understand this issue, consider the example in which the response for one chromatic direction is well into the saturated regime of the contrast-response function, while the other direction is not. This could lead to artifacts in the measured ratio of the response to the two directions with eccentricity, where any change in response for the saturated direction is hidden by a ceiling effect. Our study minimizes the role of this factor through measurement and modeling of the contrast response functions in each chromatic direction, allowing the QCM to extract a contrast independent shape for the isoresponse contours.

Although changes in sensitivity with eccentricity can be caused by mechanisms at many levels of the visual system, the lack of such variation in our data is parsimoniously explained by retinal output that preserves chromatic sensitivity with eccentricity. This interpretation is challenged by studies that show random (*Wool et al., 2018*) or close to random (*Field et al., 2010*) inputs from L- and M-cones to midget ganglion cell centers in the retinal periphery. Not all studies of midget cell chromatic

responses or their parvocelluar LGN counterparts agree with non-selective wiring (*Reid and Shapley, 1992*; *Martin et al., 2001*; *Martin et al., 2011*; *Lee et al., 2012*). One possible cortical mechanism that could compensate for the reduced L-M signal-to-noise ratio with eccentricity as predicted by random wiring models is supra-threshold compensation for reduced signals. Mechanisms of this sort have been postulated in the domain of contrast perception (*Georgeson and Sullivan, 1975*) and anomalous trichromatic vision (*Boehm et al., 2014*; *Tregillus et al., 2021*). Another possibility is a differential change in stimulus integration area across chromatic directions and with eccentricity. However, this latter possibility is not supported by fMRI population receptive field (pRF) measurements for modulations in different chromatic directions in V1 (*Welbourne et al., 2018*).

### Generalizing the QCM

In our study, we only measured responses to stimuli confined to the LM contrast plane. A more general account of chromatic contrast sensitivity requires modulating stimuli in all three-dimension of the full L-, M-, and S-cone contrast space. The QCM may be generalized in a straightforward manner to handle this expanded stimulus set by replacing the elliptical isoresponse contours with ellipsoidal isoresponse surfaces, but we have yet to test this generalization.

Another way in which the QCM may be generalized, even within the LM contrast plane, is to consider modulations at other temporal frequencies. In our experiment, we fixed the temporal modulation of the stimulus at 12 Hz. The QCM could be readily fit to data from modulations at various other temporal frequencies with the goal of observing how the chromatic sensitivity changes. With this, one could further examine the BOLD response to mixtures of different temporal frequency modulations. This is particularly interesting in that any arbitrary complex temporal modulation can be decomposed into an additive mixture of modulations at different temporal frequencies and phases (*Bracewell, 1978*). If the response to temporal frequency mixtures can be predicted via a simple rule of combination (such as linearity), establishing the QCM parameters for a well-chosen set of temporal frequencies would enable prediction of the BOLD response to chromatic stimuli modulated with complex temporal sequences.

Just as we had a fixed temporal frequency in our experiment, the stimulus presented also had a fixed spatial frequency (0 c.p.d.). Similar to the temporal domain, complex spatial images can be broken down into a combination of oriented two-dimensional sine wave patterns at single spatial frequencies and phases. Examining how the QCM fits change with changing spatial frequencies might allow for models of the BOLD response to arbitrary complex spatial stimuli. Consistent with this general goal, recent work has developed quantitative forward models of the BOLD response of early visual cortex to a variety of achromatic modulations with different spatial patterns (*Kay et al., 2013a*; *Kay et al., 2013b*). These models have sequential stages of processing that operate on an input image and transform it into a model of the BOLD response. How such models should be generalized to handle chromatic modulations is not known. If the QCM holds for other spatial frequencies, such a result would place important constraints on the appropriate generalization for such forward models to incorporate color.

## Materials and methods

### Subjects

Three subjects (age 23, 25, and 26 years; two female) took part in the fMRI experiment. All subjects had normal or corrected to normal acuity and normal color vision. The research was approved by the University of Pennsylvania Institutional Review Board. All subjects gave informed written consent and were financially compensated for their participation.

### Experimental overview

Each subject participated in four sessions of data collection. The first two sessions constituted Measurement Set 1, and the second two sessions Measurement Set 2 (Measurement Set 2 being a replication of Measurement Set 1). In both sessions, subjects underwent 48 min of fMRI scanning, with Session 1 also including two anatomical scans (a T1-weighted and a T2-weighted scan). Subjects were tested for color vision deficiencies in a separate session, using the Ishihara pseudoischromatic plates (*Ishihara, 1977*). All subjects passed with no errors. The experimental procedures for

Measurement Set 1 were preregistered (https://osf.io/wgfzy/), and an addendum describes the replication Measurement Set 2 (https://osf.io/zw6jp/).

## Digital light synthesis and silent substitution

All stimuli were generated using a digital light synthesis device (OneLight Spectra). This device produces desired spectra through the use of a digital micro-mirror device chip that in the configuration we used allows for the mixture of 56 independent primaries with a FWHM of ~16 nm and a refresh rate of 100 Hz.

Stimuli were generated to evoke specific photoreceptor responses through the use of silent substitution (*Estévez and Spekreijse, 1982*). Silent substitution operates on the principle that there exist sets of light spectra that, when exchanged, selectively modulate the activity of specified cone photoreceptors. Thus, stimulus modulations relative to a background can be generated such that they nominally modulate the activity of only the L-, M-, or S-cones, or combinations of cone classes at specified contrasts. Additional information on how the stimuli were generated is provided in *Spitschan et al., 2015*.

The stimuli account for differences in the cone fundamentals between the fovea and the periphery. This was done by treating the L, M, and S cones in the 2- and 15-degree CIE physiologically-based cone fundamentals (*CIE, 2007*) as six classes of photoreceptors. For any desired set of L-, M-, and S-cone contrasts, we designed modulations that attempted to provide the same contrasts on the 2- and 15-degree L-cone fundamentals, on the 2- and 15-degree M-cone fundamentals, and on the 2- and 15-degree S-cone fundamentals. This is possible because our device has 56 primaries, rather than the typical 3 of RGB displays. Our procedure has the effect of creating light spectra that reduce differences in the L-, M-, and S-cone contrasts produced across the retina. The cone fundamentals were tailored to the age of each subject, to account for age-related differences in typical lens density (*CIE, 2007*). See *Tables 5–7* for the central and peripheral maximum stimulus contrast values for each subject and measurement set.

Spectroradiometric measurements of the stimuli were made before and after each experimental session. During the measurements made prior to the experiment, a correction procedure was run in which the spectral power distribution of the modulation in each chromatic direction were adjusted to minimize the difference between the measured and desired cone contrasts for the 2- and 15-degree cone fundamentals. These corrections were made to the modulation spectra at the maximum contrast used in each direction. The order in which spectroradiometric measurements were taken during an experimental session was (1) five pre-correction measurements for each chromatic direction used in the session, (2) the corrections procedure, (3) five post-correction measurements per direction, and (4) five post-experiment measurements per direction. The mean of post-correction and post-experiment cone contrast measurements for the individual subjects and measurement sets are provided in *Tables 5–7*.

## Visual stimuli

The stimuli were confined to the LM plane of cone contrast space (*Figure 10A*; see also *Figure 1A*). Cone contrast space has three axes that are defined by the relative change in the quantal catch of the L, M, and S cones when modulating between the light spectra of interest and a specified reference spectrum. We refer to the reference spectrum used to calculate this relative change in cone excitations as the background (nominal chromaticity; x = 0.459, y = 0.478, luminance Y = 426 cd/m2; chromaticity and luminance computed with respect to the XYZ 10° physiologically-relevant color matching functions, as provided at https://cvrl.org). The background corresponds to the origin of cone contrast space. The LM contrast plane is a subspace of cone contrast space consisting of modulations that affect only L- and M-cone excitations, but which leave S-cone excitations unchanged relative to the background. A point in cone contrast space specifies how much L- and M-cone contrast is produced by modulating from the background to the specified stimulus. Points lying along the x-axis of *Figure 10A* modulate only L-cone contrast while M-cone contrast remains constant. Points lying along the y-axis modulate only M-cone contrast while keeping L-cone contrast constant. Points in intermediate directions modulate both L- and M-cone contrast, in proportion to the x- and y-axis components.

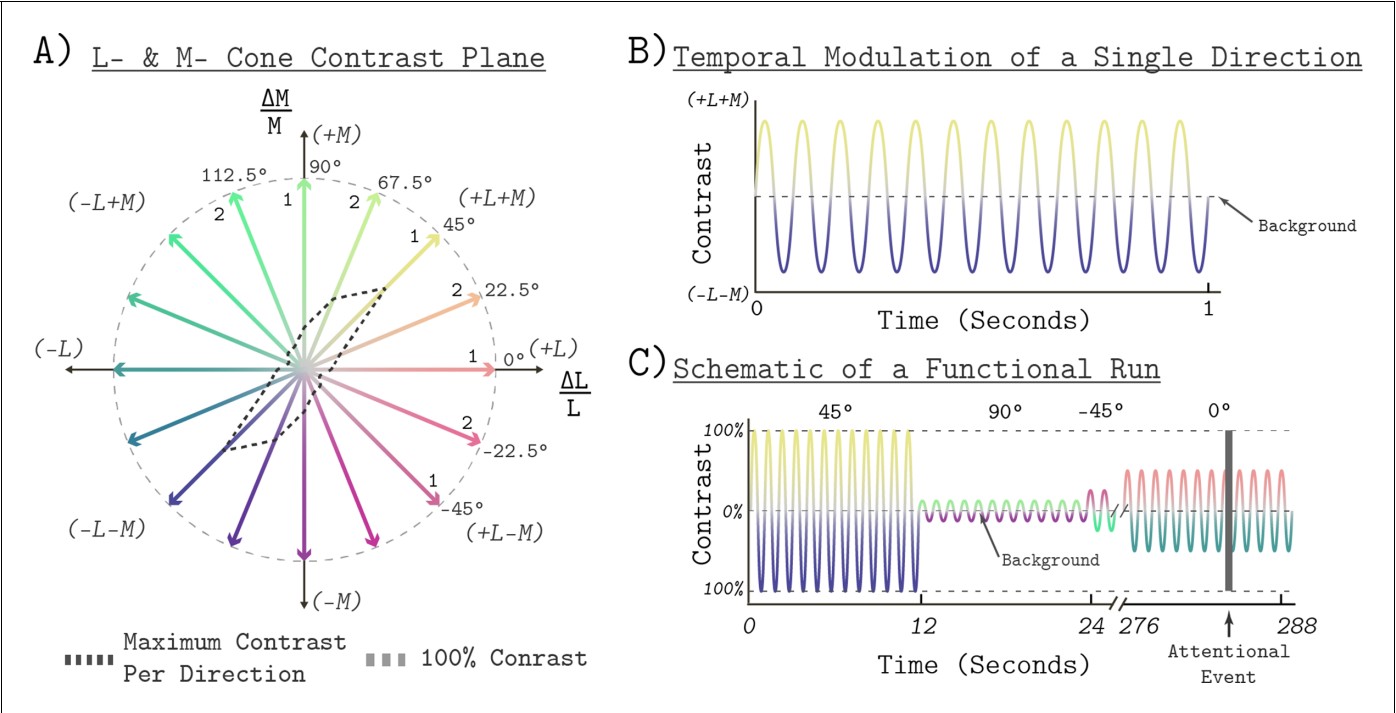

**Figure 10.** Stimulus space and temporal modulations. (**A**) The LM contrast plane. A two-dimensional composed of axes that represent the change in L- and M-cone activity relative to the background cone activation, in units of cone contrast. Each aligned pair of vectors in this space represents the positive (increased activation) and negative (decreased activation) arms of the bipolar temporal modulations. We refer to each modulation by the angle of the positive arm in the LM contrast plane, with positive ΔL/L being at 0°. The black dashed lines show the maximum contrast used in each direction. The gray dashed circle shows 100% contrast. The '1' or '2' next to each positive arm denotes the session in which a given direction presented. The grouping was the same for Measurement Set 1 and Measurement Set 2. (**B**) The temporal profile of a single bipolar chromatic modulation. This shows how the cone contrast of a stimulus changed over time between the positive and negative arms for a given chromatic direction. The particular direction plotted corresponds to the 45° modulation at 12 Hz temporal frequency. The temporal profile was the same for all chromatic directions. (**C**) Schematic of the block structure of an functional run. Blocks lasted 12 s and all blocks were modulated around the same background. The amplitude of the modulation represents the contrast scaling, relative to its maximum contrast, for that block. Each run lasted a total of 288 s. The dark gray vertical bar represents an attentional event in which the light stimulus was dimmed for 500 ms.

All stimuli were spatially uniform (0 c.p.d.), full-field temporal modulations with a radius of 30° visual angle. The temporal modulations were bipolar sinusoids around the reference background, with the positive and negative arms designed to increase and decrease targeted cone excitations in a symmetric fashion. All stimuli were modulated at 12 Hz (*Figure 10B*). A single modulation is thus described by pair of vectors in the LM contrast plane that have an angle of 180° between them, corresponding to the positive and negative arms of the modulation (*Figure 10A*). The entries of the vectors are the L- and M-cone contrasts of the end points of each arm. In this paper, we refer to a modulation by the angle made between its positive arm and the positive x-axis (corresponding to 0° in the LM contrast plane), with angle increasing counterclockwise. We refer to each angle tested as a chromatic direction. In total, we tested the eight chromatic directions: −45°, −22.5°, 0°, 22.5°, 45°, 67.5°, 90°, and 112.5°. The −45°, 0°, 45°, and 90° directions correspond to L-M, L-cone isolating, L+M, and M-cone isolating directions, respectively. For all chromatic directions, the spectra were designed to produce constant S-cone contrast.

We express the stimulus contrast of a modulation as the vector length (L2 norm) of the cone contrast representation of its positive arm (*Figure 10A*). Gamut limitations of the light synthesis engine result in different maximum stimulus contrasts for different chromatic directions (see heavy dashed contour in *Figure 10A*). The maximum contrast used in each direction is provided in *Table 3*. For all directions, we tested five contrast levels. The contrast levels tested for each chromatic direction were selected to be log spaced relative to the maximum contrast used. The relative contrasts were

**Table 3.** Table of the nominal maximum contrast per direction.
The top row indicates the chromatic direction in the LM plane.The L, M, and S contrast rows show the desired contrast on the L, M, and S cones, respectively. The total contrast is the vectorlength of stimuli made up of the L, M, and S cone contrast components and is the definition of contrast used in this study.

| Direction | −45° | −22.5° | 0° | 22.5° | 45° | 67.5° | 90° | 112.5° |
|---|---|---|---|---|---|---|---|---|
| L-Contrast | 8.49% | 7.85% | 14% | 18.48% | 42.43% | 15.31% | 0% | 4.98% |
| M-Contrast | 8.49% | 3.25% | 0% | 7.65% | 42.43% | 36.96% | 22% | 12.01% |
| S-Contrast | 0% | 0% | 0% | 0% | 0% | 0% | 0% | 0% |
| Total Contrast | 12% | 8.5% | 14% | 20% | 60% | 40% | 22% | 13% |

100, 50, 25, 12.5, and 6.25 percent. We also measured a 0 contrast reference condition in which the background without modulation was presented.

## Experimental design

We measured whole brain BOLD fMRI responses to stimuli modulated in eight different chromatic directions, each with five contrast levels, using a block design. In total, we tested 40 different combinations of contrast levels and chromatic directions. We split the eight chromatic directions into two separate scanning sessions of four directions each (*Figure 11A*). In Session 1, we tested −45° (L-M), 0° (L Isolating), 45° (L+M), and 90° (M Isolating). In Session 2, we tested the other four directions (−22.5°, 22.5°, 67.5°, and 112.5°). The order of data collection for the two sessions was randomized across subjects. A session consisted of 10 runs and each run had a duration of 288 s. Within a run, each contrast/direction pair was presented within 12 s blocks (*Figure 11*). The order of contrast/direction pairs was psuedorandomized within each run. Along with four presentations of a background-only block, each run consisted of 24 blocks. The background-only blocks contained no temporal contrast modulation, providing a reference condition for data analysis. We chose 12 Hz modulations based upon prior work showing that for stimuli similar to the ones used in this study (L+M and L-M), this frequency elicited a robust response in V1 (*Spitschan et al., 2016*). Within a block, modulations were ramped on and off using a 500 ms half-cosine. *Figure 10C* provides a schematic of the structure of an functional run.

During each functional run, subjects engaged in an attention task. This task consisted of pressing a button every time the stimulus dimmed (*Figure 10C*). Each attentional event lasted for 500 ms. The probability of an attentional event occurring in a block was 66% in Measurement Set 1 and a 33% in Measurement Set 2. The onset time of an attentional event within a block was random except that the event could not occur during the on and off half-cosine ramp. The purpose of the attention task was to encourage and monitor subject wakefulness throughout the scan session. All subjects responded to 100% of the attentional events throughout all runs and sessions.

Each subject was studied during an initial pair of scanning sessions that we call Measurement Set 1 (*Figure 11A*), and a subsequent pair of replication scans that we call Measurement Set 2 (*Figure 11B*). Measurement Set 2 tested stimuli with the same chromatic directions and contrast levels as Measurement Set 1. The grouping of chromatic directions within a session was the same across measurement sets. The two measurement sets used different pseudo-randomized presentation orders. Both measurement sets also randomly assigned which session was acquired first. Across both sessions and measurements sets, we collected a total of 960 blocks per subject. The two measurement sets were analyzed separately.

## MRI data acquisition

MRI scans made use of the Human Connectome Project LifeSpan protocol (VD13D) implemented on a 3-Tesla Siemens Prisma with a 64-channel Siemens head coil. A T1-weighted, 3D, magnetization-prepared rapid gradient-echo (MPRAGE) anatomical image was acquired for each subject in axial orientation with 0.8 mm isotropic voxels, repetition time (TR) = 2.4 s, echo time (TE) = 2.22 ms, inversion time (TI) = 1000 ms, field of view (FoV) = 256 mm, flip angle = 8°. BOLD fMRI data were obtained over 72 axial slices with 2 mm isotropic voxels with multi-band = 8, TR = 800 ms, TE = 37 ms, FOV = 208 mm, flip angle = 52°. Head motion was minimized with foam padding.

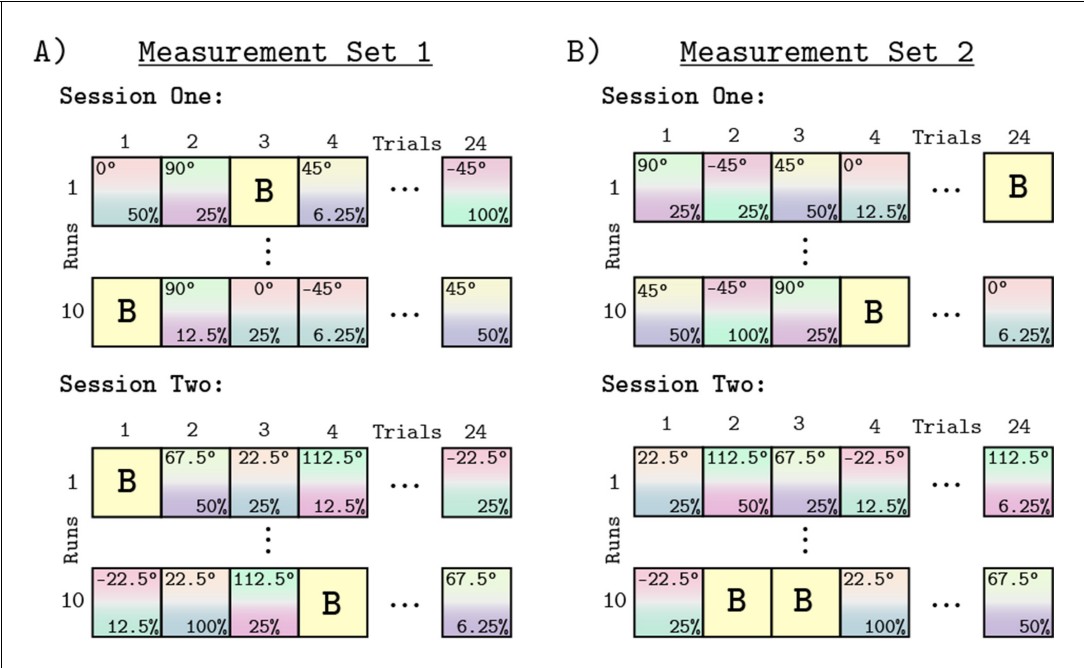

**Figure 11.** Experimental design. Panels **A** and **B** show the block design used for all runs and sessions. Panel **A** shows Measurement Set 1 which contained two separate MRI sessions. Each session contained four of the eight chromatic directions. The split of directions across the two sessions was the same for all subjects, but which session each subject started with was randomized. Within a session we collected 10 functional runs, each containing 24 blocks. The 24 blocks consisted of 20 direction/contrast paired stimulus blocks (depicted by the gradient squares with direction noted at top and the contrast at bottom of each square) and 4 background blocks (squares marked 'B'). The order of blocks within each run was randomized, with each contrast/direction pair shown once per run. Each run had a duration of 288 second. Panel **B** shows Measurement Set 2, which was a replication of Measurement Set 1, with session order and order of blocks within run re-randomized. There were 960 blocks across both measurement sets.

During MRI scanning, subjects were supine inside the magnet. Visual stimuli were presented through an MRI compatible eyepiece to the right eye of the subject. Stimuli were delivered from the digital light synthesizer through a randomized optical fiber cable (Fiberoptics Technology Inc). The randomization of the fiber optic cable helped to minimize any spatial inhomogeneities in the spectrum of the stimulus. The eye piece provided adjustable focus to account for variation in refractive error. As the stimulus was a spatially uniform field, however, the effect of any spatial blur upon the stimulus was minimal.

Subjects used either button of a two button MR compatible response device (Current Designs) to respond to attention events during the functional runs.

## MRI data preprocessing

Both anatomical and functional data were preprocessed according to the HCP minimal preprocessing pipeline (*Glasser et al., 2013*) Briefly, the anatomical data were passed through the pre-freesurfer, freesurfer, and post-freesurfer steps of the HCP minimal preprocessing pipeline. This was used to create an MNI registration, a Freesurfer segmentation (*Dale et al., 1999*; *Fischl et al., 1999*), and a surface mesh. The functional data were preprocessed with both the volume and surface pipelines. The volume pipeline applied gradient distortion correction, motion correction (FLIRT 6 DoF; *Jenkinson et al., 2002*), top-up phase encoding distortion correction (*Smith et al., 2004*), and registered the functional images to the anatomical images. The surface pipeline mapped the volume data to the CIFTI grayordinate space which includes a left and right 32K cortical surface mesh and subcortical voxels. Finally, the functional data were passed through the ICAFIX pipeline, which uses independent component analysis and noise/not-noise classification to denoise the time course.

After initial preprocessing, we performed a series of subsequent steps before analyzing the time course data. We used a V1 region of interest (ROI) to extract the time series from primary visual cortex (see below for definition of retinotopic maps). The signals from each voxel were mean centered

**Table 4.** Number of censored fMRI frames per run.

Values shown are for Subject 1 Measurement Set 2. The top set of rows show data for session one and the bottom set of the show data for session 2. Each set of rows show the number of censored frames per run out of 360 frames. Subjects and sessions not shown mean that no frames were censored in those runs.

**Subject 1 – Measurement Set 2: Session 1**

| Run Number | 1 | 2 | 3 | 4 | 5 | 6 | 7 | 8 | 9 | 10 |
|---|---|---|---|---|---|---|---|---|---|---|
| Number of Censored Frames (n/360) | 0 | 0 | 8 | 18 | 0 | 26 | 47 | 4 | 0 | 0 |

**Subject 1 – Measurement Set 2: Session 2**

| Run Number | 1 | 2 | 3 | 4 | 5 | 6 | 7 | 8 | 9 | 10 |
|---|---|---|---|---|---|---|---|---|---|---|
| Number of Censored Frames (n/360) | 0 | 0 | 0 | 0 | 0 | 0 | 0 | 7 | 0 | 0 |

and converted to percent signal change. We then performed nuisance regression using the relative motion estimates and attentional events as regressors. The relative motion regressors (translation of X, Y, and Z and yaw, pitch, and roll) were mean centered and scaled by their respective standard deviations. The attention event regressor was modeled as a series of delta functions located within the TRs in which the events occurred, convolved with a hemodynamic response function. The nuisance regression was performed using the MATLAB linear regression function *mldivide* (MathWorks) with the residual of the model used as the 'cleaned' timed series.

Next, we time-point censored the time series of all voxels based on the motion estimates. This was done using a modified version of *Power et al., 2014*. We converted the relative yaw, pitch, and roll estimates to millimeters by using the distance that each of these angles subtend on an assumed 50 mm sphere. We then took the L2-norm of the six translation and mm rotation estimates as a metric of frame-wise displacement (FD). We censored three contiguous TRs centered on any time point with an FD > 0.5. Time points that exceeded the threshold were excluded from analysis, and a table of the number of censored frames can be found in *Table 4*. Finally, we applied polynomial detrending by fitting a fifth order polynomial to the time course from each voxel and subtracting it from the signal. Analyses performed at the level of V1 were done using the median value across voxels at each time point to represent the V1 signal. Vertex-wise analyses were performed on the preprocessed time course of individual vertices.

## Definition of retinotopic maps

Retinotopic regions of interest (ROI) were defined using the anatomical template neuropythy (*Benson et al., 2014*) which provides eccentricity and polar angle maps for V1, V2, and V3. From this atlas, we defined a V1 ROI using the voxels in area V1 between 0–20° eccentricity and 0–180° polar angle. We set the eccentricity upper bound of the ROI to be 20° to provide a conservative boundary to ensure that we only analyzed stimulated vertices. This accounts for some uncertainty in the exact retinal size of the stimulated area due to, for example, variation in the distance of the eyepiece to the eye of the subject.

To define the retinotopic regions of interest used for hV4/Vo1 analysis, we registered the *Wang et al., 2015* retinotopic atlas to CIFTI space. This was achieved using the subject specific atlas as defined by Neuropythy (*Benson and Winawer, 2018*). This atlas was then registered to MNI space through the use of ANTs (*Avants et al., 2008*). With the atlas in MNI space, we then transformed it to CITFI greyordinate space and used the hV4 and the VO1 ROIs to extract the time series data.

## Subject-specific hemodynamic response function

We derived subject-specific hemodynamic response functions (HRFs) for each subject and measurement set. The HRF was derived using all the functional runs within a measurement set, using the V1 region of interest median data. The time-series data were fit with a non-linear model that simultaneously estimated the beta weights of the GLM to fit stimulus responses, and the parameters of an HRF model. The HRF model was composed of the first three components of the 'FLOBS' basis set (*Woolrich et al., 2004*). The best fitting HRF model was then used to fit that subject's data for both the GLM and QCM models.

**Table 5.** Stimulus validation measurements for Subject 1.

The top set of rows show data for Measurement Set 1 and the bottom set show data for Measurement Set 2. The dark gray rows show the nominal angle and contrast. Each cell shows the mean and standard deviation of stimulus vector angles and lengths computed from 10 validation measurements (5 pre-experiment, 5 post-experiment). Center and periphery denote which set of cone fundamentals were used to calculate cone contrast of the stimuli referring either the 2° or 15° CIE fundamentals, respectively.

**Subject 1 – Measurement Set 1**

| Nominal Angle | −45° | −22.5° | 0° | 22.5° | 45° | 67.5° | 90° | 112.5° |
|---|---|---|---|---|---|---|---|---|
| Center Angle | −41.23 ±3.13 | −15.90 ±6.59 | 2.17 ±1.05 | 23.43 ±0.23 | 45.21 ±0.23 | 69.36 ±1.61 | 87.98 ±1.26 | 120.94 ±8.45 |
| Periphery Angle | −42.85 ±3.04 | −16.31 ±6.18 | 1.61 ±0.44 | 22.75 ±0.22 | 44.78 ±0.22 | 68.13 ±1.54 | 88.72 ±0.18 | 119.67 ±8.04 |
| **Nominal Contrast** | **12%** | **8.5%** | **14%** | **20%** | **60%** | **40%** | **22%** | **13%** |
| Center Contrast | 12.14 ±0.05 | 9.02 ±0.75 | 14.44 ±0.37 | 21.05 ±0.79 | 60.92 ±0.09 | 38.01 ±1.97 | 21.39 ±0.85 | 12.56 ±0.38 |
| Periphery Contrast | 12.00 ±0.04 | 8.93 ±0.72 | 13.98 ±0.35 | 20.28 ±0.73 | 58.73 ±0.09 | 37.81 ±1.89 | 21.42 ±0.55 | 12.48 ±0.36 |

**Subject 1 - Measurement Set 2**

| Nominal Angle | −45° | −22.5° | 0° | 22.5° | 45° | 67.5° | 90° | 112.5° |
|---|---|---|---|---|---|---|---|---|
| Center Angle | −45.09 ±0.55 | −22.44 ±0.98 | 3.49 ±2.27 | 22.97 ±0.15 | 45.18 ±0.03 | 67.75 ±0.24 | 87.81 ±1.69 | 112.99 ±1.03 |
| Periphery Angle | −46.16 ±0.55 | −22.32 ±0.99 | 3.24 ±2.77 | 22.39 ±0.19 | 44.92 ±0.02 | 66.85 ±0.21 | 87.33 ±1.24 | 68.34 ±0.92 |
| **Nominal Contrast** | **12%** | **8.5%** | **14%** | **20%** | **60%** | **40%** | **22%** | **13%** |
| Center Contrast | 11.75 ±0.01 | 8.28 ±0.08 | 13.78 ±0.09 | 20.35 ±0.22 | 60.26 ±0.48 | 37.62 ±0.11 | 21.82 ±0.08 | 12.69 ±0.05 |
| Periphery Contrast | 11.65 ±0.02 | 8.29 ±0.08 | 13.53 ±0.09 | 19.79 ±0.18 | 58.47 ±0.48 | 38.16 ±0.10 | 21.99 ±0.10 | 12.81 ±0.06 |

## General linear model

We used an ordinary least squares regression with a stimulus design matrix that described the stimulus order of a run. The regression matrix contained one regressor per stimulus block as well as a single regressor for the baseline, with the length of the regressor equal to the number of timepoints (360 TRs). The regressor for each stimulus block in a run was modeled by a binary vector that indicated the timepoints when the stimulus was present (ones) or absent (zeros), convolved with the HRF. This resulted in 21 GLM beta weights per run. For all analyses, model fitting was performed using the concatenation of all functional runs within a measurement set. For fitting of the GLM, we concatenated the stimulus design matrices such that like contrast/direction pairs were modeled by the same regressor. The concatenated stimulus design matrix for a measurement set had a total of 41 regressors (20 direction/contrast pairs from Session 1, 20 direction/contrast pairs from Session 2, and a shared baseline regressor) and 7200 timepoints. Additional details of the GLM are provided in the Appendix 1 section.

## Contrast response functions

To obtain contrast response function for each color direction, the time course data for each run was fit using a general linear model to obtain the effect that each stimulus had on the BOLD fMRI response. Grouping the GLM beta weights by chromatic direction defined a set of 8 contrast response functions (CRFs), one per direction. A CRF describes the relationship, within a particular chromatic direction, between the contrast of the stimulus and the measured response. The CRFs obtained using the GLM beta weights fit to the concatenated time series of Subject 2 can be seen in *Figure 3*. The panels of *Figure 3* show the CRFs for the eight different chromatic directions. The CRFs for Subject 1 and 2 can be seen in *Figure 2—figure supplements 1–2*.

**Table 6.** Stimulus validation measurements for Subject 2.

The format of this table is the same as *Table 5*. Cells that contain an 'X' mark stimulus directions in which validation measurements were not recorded due to technical difficulty.

**Subject 2 - Measurement Set 1**

| Nominal Angle | −45° | −22.5° | 0° | 22.5° | 45° | 67.5° | 90° | 112.5° |
|---|---|---|---|---|---|---|---|---|
| Center Angle | −45.73 ±1.06 | −19.23 ±2.51 | 1.71 ±0.98 | 24.04 ±0.39 | 45.43 ±0.09 | 68.36 ±0.53 | 87.87 ±1.64 | 114.92 ±1.90 |
| Periphery Angle | −47.03 ±1.02 | −18.91 ±2.40 | 1.63 ±0.77 | 23.65 ±0.38 | 44.99 ±0.09 | 67.18 ±0.52 | 87.65 ±1.09 | 114.11 ±1.86 |
| **Nominal Contrast** | **12%** | **8.5%** | **14%** | **20%** | **60%** | **40%** | **22%** | **13%** |
| Center Contrast | 12.04 ±0.11 | 8.41 ±0.21 | 14.01 ±0.08 | 20.45 ±0.52 | 60.72 ±0.38 | 39.68 ±0.63 | 21.82 ±0.23 | 12.73 ±0.16 |
| Periphery Contrast | 11.88 ±0.11 | 8.36 ±0.21 | 13.58 ±0.07 | 19.81 ±0.51 | 58.56 ±0.35 | 39.41 ±0.59 | 21.82 ±0.24 | 12.62 ±0.14 |

**Subject 2 - Measurement Set 2**

| Nominal Angle | −45° | −22.5° | 0° | 22.5° | 45° | 67.5° | 90° | 112.5° |
|---|---|---|---|---|---|---|---|---|
| Center Angle | X | −25.85 ±3.20 | X | 22.72 ±0.38 | X | 67.42 ±0.16 | X | 112.03 ±1.22 |
| Periphery Angle | X | −26.43 ±3.04 | X | 22.35 ±0.35 | X | 66.36 ±0.16 | X | 110.12 ±1.17 |
| **Nominal Contrast** | **12%** | **8.5%** | **14%** | **20%** | **60%** | **40%** | **22%** | **13%** |
| Center Contrast | X | 8.36 ±0.16 | X | 19.77 ±0.23 | X | 39.75 ±0.13 | X | 12.75 ±0.15 |
| Contrast | X | 8.34 ±0.17 | X | 19.21 ±0.23 | X | 40.04 ±0.15 | X | 12.95 ±0.13 |

## Error bars

The error bars and error regions in all figures are the 68 percent confidence intervals. We used 68% confidence intervals as this approximates +/- 1 SEM for a normal distribution. The percentiles we use to estimate error are the result of a bootstrap analysis. The bootstrap analysis was implemented as random sampling with replacement of the runs within a measurement set. The randomly drawn runs in both sessions were concatenated and fit by all models. We performed 200 bootstrap iterations and identified the 68% percent confidence interval from the bootstrap results.

## Leave-runs-out cross-validation

To evaluate model performance, we employed a leave-runs-out cross-validation strategy. For each cross-validation iteration, runs from Session 1 and Session 2 were randomly paired within the same measurement set. These pairs of runs were held out and the models were fit to the remaining 18 runs. From these model fits, a time course prediction for the left-out runs were obtained from both models. We computed the R2 value between these predictions and the time course of the held-out runs. The average R2 value across the 10 cross-validation iterations was used to compare models.

## Leave-session-out cross-validation

To evaluate the generalizability of the QCM, we implemented a leave-session-out cross-validation. As the eight chromatic directions tested were separated into two sessions with the same grouping across all subjects and measurement sets, we could evaluate the ability of the QCM to make predictions for chromatic directions and contrasts not used to fit the model parameters. Within each measurement set, we fit the QCM to Sessions 1 and 2 separately and evaluated how well the parameters of the model predicted responses to stimulus directions in the held-out session. These predicted responses were grouped by chromatic direction in order to construct a set of contrast response functions. The error bars in the CRFs are the 68% confidence intervals computed using bootstrapping, where we randomly sampled runs with replacement and compute the leave-session-out cross-validation a total a 200 times.

**Table 7.** Stimulus validation measurements for Subject 3.
The format of this table is the same as *Tables 5,6*. Cells that contain an 'X' mark stimulus directions in which validation measurements were not recorded due to technical difficulty.

**Subject 3 - Measurement Set 1**

| Nominal Angle | −45° | −22.5° | 0° | 22.5° | 45° | 67.5° | 90° | 112.5° |
|---|---|---|---|---|---|---|---|---|
| Center Angle | −48.84 ±5.02 | −27.84 ±6.02 | 5.29 ±4.45 | 23.61 ±0.77 | 45.41 ±0.15 | 67.48 ±0.42 | 85.96 ±3.11 | 108.56 ±4.89 |
| Periphery Angle | −50.40 ± 4.90 | −28.48 ±5.97 | 5.09 ±4.72 | 23.26 ±0.77 | 45.02 ±0.13 | 66.33 ±0.43 | 86.12 ±3.24 | 107.49 ±4.78 |
| **Nominal Contrast** | **12%** | **8.5%** | **14%** | **20%** | **60%** | **40%** | **22%** | **13%** |
| Center Contrast | 12.25 ±0.32 | 8.39 ±0.06 | 13.96 ±0.08 | 19.78 ±0.20 | 61.92 ±1.40 | 40.87 ±1.14 | 22.91 ±1.30 | 13.53 ±0.75 |
| Periphery Contrast | 12.13 ±0.31 | 8.30 ±0.08 | 13.52 ±0.09 | 19.17 ±0.21 | 59.76 ±1.36 | 40.65 ±1.13 | 22.97 ±1.26 | 13.47 ±0.71 |

**Subject 3 - Measurement Set 2**

| Nominal Angle | −45° | −22.5° | 0° | 22.5° | 45° | 67.5° | 90° | 112.5° |
|---|---|---|---|---|---|---|---|---|
| Center Angle | X | −24.51 ±1.57 | X | 22.92 ±0.07 | X | 67.65 ±0.19 | X | 111.23 ±1.11 |
| Periphery Angle | X | −24.17 ±1.53 | X | 22.65 ±0.06 | X | 66.57 ±0.19 | X | 69.95 ±1.07 |
| **Nominal Contrast** | **12%** | **8.5%** | **14%** | **20%** | **60%** | **40%** | **22%** | **13%** |
| Center Contrast | X | 8.24 ±0.07 | X | 19.93 ±0.09 | X | 39.77 ±.0.27 | X | 12.88 ±0.11 |
| Periphery Contrast | X | 8.23 ±0.07 | X | 19.45 ±0.26 | X | 40.16 ±0.26 | X | 12.92 ±0.10 |

## Surface parameter map generation

Cortical surface maps were generated to visualize the ellipse angle and minor axis ratio parameters of the QCM on the V1 cortical surface. To generate the surface parameter maps, we fit the QCM, within a single vertex, to the concatenated time course from all runs in a measurement set. We repeat this fit for all vertices within the visual areas template map from neuropythy (*Benson et al., 2014*). To visualize the surface parameter maps in a manner that highlights differences in fits as a function of cortical position, we created a series of scatter plots that relate the minor axis ratio and angle parameters to the eccentricity of their respective vertices. The regression lines in the scatter plots are robust regression lines implemented through the built in MATLAB function *robustfit* which adaptively reweights the data to discount the effects of outliers.

## Parameter fitting

We fit QCM the model to the concatenated time series for both sessions within a measurement set for each subject. The data were fit using the MATLAB function *fmincon* to find a set of model parameters that minimize the difference between the actual fMRI time course and the QCM prediction of the time course, computed as the root mean squared error.

## Preprocessing and analysis code

All code used to perform analyses in this paper may be found in our public GitHub repository: https://github.com/BrainardLab/LFContrastAnalysis_eLife.git (copy archived at swh:1:rev: c9b4cbe72e69e4d3d623ec8b7fc62076e2fe1a22, *Barnett, 2021*).

## Spectroradiometric stimulus validations

*Tables 5– 7* show the stimulus validation measurements for all subjects and sessions. The tables show the mean and standard deviation of stimulus vector angles and lengths computed from 10 validation measurements (five pre-experiment, five post-experiment). Center and periphery denote

which set of cone fundamentals were used to calculate cone contrast of the stimuli referring either the 2° or 15° CIE fundamentals, respectively.

## Acknowledgements

This work has been supported by the National Institutes of Health (Grants:RO1 EY10016 and Core GrantP30 EY001583; https://www.nih.gov/) and National Science Foundation Graduate Research Fellowship (DGE-1845298). We thank Joris Vincent, Jack Ryan, Ozenc Taskin, and Nicolas Cottaris for technical assistance.

## Additional information

### Funding

| Funder | Grant reference number | Author |
|---|---|---|
| National Science Foundation | DGE-1845298 | Michael A Barnett |
| National Institutes of Health | RO1 EY10016 | David Brainard |
| National Institutes of Health | P30 EY001583 | David Brainard |

The funders had no role in study design, data collection and interpretation, or the decision to submit the work for publication.

### Author contributions

Michael A Barnett, Software, Formal analysis, Funding acquisition, Validation, Investigation, Visualization, Methodology, Writing - original draft, Project administration, Writing - review and editing; Geoffrey K Aguirre, Conceptualization, Methodology, Writing - review and editing; David Brainard, Conceptualization, Software, Formal analysis, Funding acquisition, Methodology, Writing - original draft, Writing - review and editing

### Author ORCIDs

Michael A Barnett (ID) https://orcid.org/0000-0002-8355-4601
Geoffrey K Aguirre (ID) https://orcid.org/0000-0002-4028-3100
David Brainard (ID) https://orcid.org/0000-0001-9827-543X

### Ethics

Human subjects: The research was approved by the University of Pennsylvania Institutional Review Board (Protocol: Photoreceptor directed light modulation 817774). All subjects gave informed written consent and were financially compensated for their participation.

### Decision letter and Author response

Decision letter https://doi.org/10.7554/eLife.65590.sa1
Author response https://doi.org/10.7554/eLife.65590.sa2

## Additional files

### Supplementary files

• Transparent reporting form

### Data availability

The raw fMRI data from our experiment have been deposited to OpenNeuro, under the https://doi.org/10.18112/openneuro.ds003752.v1.0.0.

The following dataset was generated:

| | Database and |
|---|---|

| Author(s) | Year | Dataset title | Dataset URL | Identifier |
|---|---|---|---|---|
| Barnett M, Aguirre G, Brainard D | 2021 | LFContrast | https://doi.org/10.18112/openneuro.ds003752.v1.0.0 | OpenNeuro, 10.18112/openneuro.ds003752.v1.0.0 |

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

## Appendix 1

### General linear model

The model used to provide a benchmark for the quadratic color model (QCM) is the general linear model (GLM). The GLM has the form:

$$Y = X\beta + \epsilon$$

This states that the measurement ($Y$) is equal to the model matrix ($X$) times the weights ($\beta$) plus the residual error ($\epsilon$). For our GLM, $Y$ is a column vector of the concatenated time series from all the fMRI runs within a measurement set (20 runs):

$$Y = \begin{bmatrix} y_1 \\ \vdots \\ y_i \\ \vdots \\ y_t \end{bmatrix}$$

The subscript $i$ indicates a particular time point, with the subscript $t$ being the total number of time points in a measurement set. Here a time point corresponds to one TR of the BOLD response, and t = 7200 (20 runs with 360 TRs). Our model comparison was for the aggregated V1 response, and we took each element $y_i$ of $Y$ to be the median BOLD fMRI response for the corresponding TR, with the median taken across the voxels in the V1 ROI.

The model matrix $X$ contains the predictor variables for the linear model. Each column of $X$ is a regressor corresponding to a single stimulus (a single chromatic direction/contrast pair or the baseline (0 contrast) uniform field). The regressors are created by convolving a binary indicator vector with the hemodynamic response function (HRF) that accounts for the sluggish BOLD response to a stimulus event. The binary indicator vectors contain 1 when the stimulus was present and 0 otherwise. The convolution then produces a predictor of the measured BOLD fMRI response for that stimulus condition. In our study, $X$ has 41 columns corresponding to the pairing of the eight chromatic directions and five contrasts levels plus the baseline

$$X = \begin{bmatrix} x_{1,1} & \cdots & x_{1,41} \\ \vdots & & \vdots \\ x_{i,1} & \cdots & x_{i,41} \\ \vdots & & \vdots \\ x_{t,1} & \cdots & x_{t,41} \end{bmatrix}$$

The general linear model predicts $Y$ as a weighted linear combination of the regressors in the columns of $X$. The weight applied to each regressor is given by the elements of $\beta$:

$$\beta = \begin{bmatrix} \beta_1 \\ \beta_2 \\ \vdots \\ \beta_{41} \end{bmatrix}$$

Using $\beta$ and $X$, we can generate a predicted time course $\hat{Y}$. We take each $\beta_i$ to be a proxy for the aggregate V1 BOLD fMRI response for the corresponding stimulus condition. This interpretation is associated with our particular choice of scaling the indicator variables in the regressors before convolution with the HRF (that is the choice of 1 for the TRs during which the stimulus was present). We determined $\beta$ using linear regression as implemented in the MATLAB mldivide operation, $\beta = X \backslash Y$ (see https://www.mathworks.com/help/matlab/ref/mldivide.html). The GLM prediction of the BOLD fMRI response at each time point $y_i$ is therefore:

$$\hat{y}_i = x_{(i,1)}\beta_1 + x_{(i,2)}\beta_2 + \ldots + x_{(i,41)}\beta_{41}$$

## Quadratic color model

The quadratic color model allows for the prediction of the BOLD fMRI response to any stimulus that lies in the LM contrast plane. This includes predictions for stimuli not in the measurement set used to fit the model. The QCM makes its predictions through three steps. The first step calculates the 'equivalent contrast' of the stimulus which can be thought of as the effective contrast of the stimulus in V1 accounting for the differences in sensitivity across chromatic directions. Equivalent contrast is the vector length of the stimuli after this linear transformation. The next step is a response non-linearity applied to the equivalent contrast to predict the underlying neural response. Finally, a convolution of this underlying response with the HRF results in predictions of the BOLD fMRI response. Here, we provide explanations and equations for the model.

We start by considering a stimulus modulation whose predicted BOLD fMRI response we wish to know. A stimulus modulation is denoted by the column vector $c$ whose two entries are the L and the M cone contrast of the stimulus ($l$ and $m$, respectively):

$$c = \begin{bmatrix} l \\ m \end{bmatrix}$$

The figure below outlines how we transform such a stimulus to a response, prior to convolution with the HRF. Panel A shows a 3-dimensional contrast-response space with the $(x,y)$ plane representing the LM contrast plane and the $z$ axis giving the response $r$ corresponding to each point in the LM contrast plane. In this representation, all possible stimuli and responses form an inverted bell shape surface, as illustrated below. At constant values of $r$ (constant height on the $z$ axis), cross sections through this surface shows the elliptical isoresponse contours of the QCM. Each elliptical isoresponse contour describes the set of LM contrast combinations that elicit the same response $r$. The teal and red dots shown in the LM plane represent example stimulus modulations, chosen in two color directions at contrasts corresponding to the five (dark blue, blue, aqua, green, yellow) isoresponse contours illustrated.

To obtain the equivalent contrast corresponding to stimulus $c$, we first apply a linear transformation $M$ to $c$ to obtain a transformed representation of the stimulus, $e = Mc$. The vector $e$ is a two-dimensional column vector whose entries we refer to as $e_1$ and $e_2$. We call the transformed representation of the stimulus the equivalent contrast space, and as we show below we choose the linear transformation $M$ such that in this space the isoresponse contours are circles. Panel B of the figure shows the same information as in Panel A represented in the equivalent contrast space. Here the $(x,y)$ plane gives the values of $e_1$ and $e_2$, and the isoresponse contours plotted with respect to the equivalent contrast plane are circular. Note that in the equivalent contrast plane, the distances between the plotted teal points are the same as the distances between the corresponding plotted red points. This is a consequence of the fact that the transformation $M$ is chosen to make the isoresponse contours circular.

A) Cone Contrast Space   B) Equivalent Contrast Space   C) Equivalent Contrast
                                                            Response Function

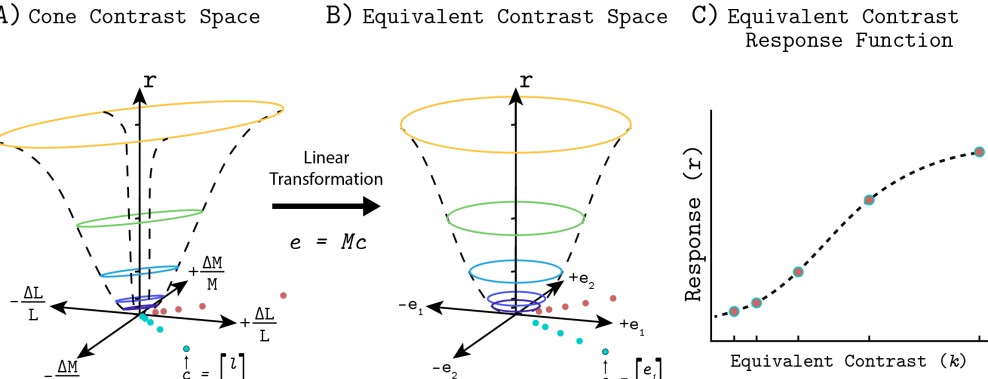

**Appendix 1—figure 1.** Illustration of cone to equivalent contrast transformation. (Panel **A**) A 3-dimensional cone contrast-response space with the $(x,y)$ plane representing the LM contrast plane and the $z$ axis giving the corresponding response $r$ to each point in the $(x,y)$ plane. The teal and red

*Appendix 1—figure 1 continued on next page*

*Appendix 1—figure 1 continued*

dots shown in the LM plane represent example stimulus modulations, chosen in two color directions at contrasts corresponding to the five isoresponse contours illustrated in dark blue, blue, aqua, green, yellow. (Panel **B**) A three-dimensional equivalent contrast-response space with the $(x,y)$ plane representing equivalent contrast ($e_1$ and $e_2$) and the $z$ axis giving the corresponding response $r$ to each point in the $(x,y)$ plane. The teal and red dots shown correspond to same color dots in Panel **A** after we apply a linear transformation $M$ to $c$ (the L- and M-cone contrast representation of the stimuli) to obtain a transformed representation of the stimulus, $e = Mc$. Note that after the application of $M$, the elliptical contours in Panel **A** are circular and the distances between the plotted teal points are the same as the distances between the corresponding plotted red points. (Panel **C**) The equivalent contrast response function. The x-axis denotes the equivalent contrast and the y-axis marks the response. The teal and red closed circles shown in Panel **C** correspond to both the teal and red points shown in Panels **A** and **B**.

We define the equivalent contrast $k$ of a stimulus as the vector length of the transformed stimulus $e$, $k = |e|$.

The next step of the model is a non-linearity that maps between equivalent contrast ($k$) and BOLD fMRI response (dashed black line in Panel C of the figure). This is possible since all stimuli in equivalent contrast space with the same equivalent contrast predict the same underlying response, regardless of the chromatic direction of the stimuli. Therefore, we can focus solely on the relationship between equivalent contrast and the associated response. We call the static non-linearity the 'equivalent contrast-response function'. The teal and red closed circles shown in Panel C correspond to both the teal and red points shown in Panels A and B, and these overlap since they lie on the same set of isoresponse contours.

The linear transformation $M$ needed to compute the equivalent contrast representation of $c$ is derived by starting with the equation for an ellipse centered at the origin, in which a positive-definite quadratic function of the cone contrasts is equal to a constant for all points on the ellipse. This equation is given below (left hand side) and re-expressed in matrix-vector form (right hand side).

$$k^2 = Al^2 + Blm + Cm^2 = \begin{bmatrix} l & m \end{bmatrix} \begin{bmatrix} A & B/2 \\ B/2 & C \end{bmatrix} \begin{bmatrix} l \\ m \end{bmatrix}$$

Changing the value of the constant $k^2$ changes the scale of the ellipse without changing its shape, and as we will see below $k$ is the equivalent contrast corresponding to each constant-shape elliptical isoresponse contour. We rewrite the matrix representation of the elliptical locus as:

$$k^2 = c^T Q c$$

This yields

$$k = \sqrt{c^T Q c}$$

Because the coefficients $A$, $B$, and $C$ are constrained so that $Q$ is a symmetric positive definite matrix, $Q$ can be factored through its eigenvalue decomposition and rewritten as

$$Q = V\Lambda V^T$$

where $V$ is an orthonormal (rotation) matrix and $\Lambda$ is a diagonal matrix with positive entries. Since $\Lambda$ is a diagonal matrix we can further decompose as

$$Q = VSS^T V^T$$

where $S$ is diagonal with entries equal to the square root of the corresponding entries of $\Lambda$.

The matrix $V$ expresses a rotation in the cone contrast plane and may be parameterized by the rotation angle $p_1$:

$$V(p_1) = \begin{bmatrix} cos(p_1) & -sin(p_1) \\ sin(p_1) & cos(p_1) \end{bmatrix}$$

The matrix $S$ is diagonal and when applied to a vector simply scales the entries of that vector.

Although $S$ normally has two degrees of freedom in the general case, we lock the scale of the major axis to 1. Therefore we are only concerned with the scaling of the minor axis ($p_2$). Thus we can define

$$S(p_2) = \begin{bmatrix} 1 & 0 \\ 0 & 1/p_2 \end{bmatrix}, 0 < p_2 \leq 1$$

In this formulation, $p_1$ and $p_2$ parameterize the shape of an elliptical isoresponse contour and $k$ parameterizes its scale. The parameter $p_1$ is what we call the angle of the major axis in the main text, while $p_2$ is the minor axis ratio.

We set $M = S^T V^T$ and rewrite $Q$ as:

$$Q = M^T M$$

The matrix $M$ then transforms the cone contrast vector $c$ to the equivalent contrast vector through $e = Mc$. Recall that equivalent contrast is the vector length of $e$. Therefore, the equivalent contrast of the points on the ellipse corresponding to $k$ is given by

$$||e|| = k$$

To see this, note that

$$||e||^2 = e^T e = c^T M^T Mc = c^T Qc = k^2$$

Thus, given $Q$, we can compute the equivalent contrast for any stimulus $c$ and apply the equivalent contrast-response function to predict the its response. The specific non-linearity we use for the contrast-response function is a Naka-Rushton function. This function is a four parameter saturating non-linearity that is defined by the following:

$$r(k) = a \frac{k^n}{k^n + s^n} + h$$

The neural response ($r$) is a function of the equivalent contrast ($k$). The parameters of the Naka-Rushton function are the amplitude ($a$), exponent ($n$), semi-saturation ($s$), and the offset ($h$). These parameters control the gain, the slope, the position along the x axis, and the y-axis offset of the non-linearity, respectively. The shape of the non-linearity controls how the underlying response changes with equivalent contrast.

Finally, we need to convert the neural response to a prediction of the BOLD fMRI response($\hat{Y}$). To do this, we convolve the underlying response with the hemodynamic response function (HRF).

$$\hat{Y} = r \circledast HRF$$

Predictions of the BOLD fMRI response via the QCM are thus made using six parameters: angle ($p_1$), minor axis ratio ($p_2$), amplitude ($p_3 = a$), exponent ($p_4 = n$), semi-saturation ($p_5 = s$), and offset ($p_6 = h$). We can define a parameter vector $P$ for the QCM as:

$$P = \begin{bmatrix} p_1 \\ \vdots \\ p_6 \end{bmatrix}$$

We fit $P$ to the data by using a non-linear parameter search routine fmincon (Matlab, see https://www.mathworks.com/help/optim/ug/fmincon.html) to find the parameter vector that that minimizes the difference between the measured BOLD fMRI time course $Y$ and the prediction $\hat{Y}$ obtained using the QCM. This takes the form of:

$$P^* = \arg\min_{P} \sqrt{\frac{\sum_{i=1}^{t} (\hat{y}_i - y_i)^2}{t}}$$

where the objective function we are minimizing is the root mean squared error. Once the values of

$P^*$ are found for an individual subject, we can use them to predict the BOLD fMRI response to any $c$ within the LM cone contrast plane.

## Linear channels model

The linear channels model (LCM) is based on the work of *Brouwer and Heeger, 2009*. They develop a model that decodes the hue angle of a stimulus a subject was viewing from BOLD fMRI measurements. The stimuli in their study were modulations in the isoluminant plane of the CIELAB color space, and were parameterized by hue angle within this plane. Their model is based on linear channels tuned to the this angular representation, and it did not explicitly consider the effect of stimulus contrast. This was sufficient for their analysis, as they studied only a single contrast for each angle. To develop a version of the LCM that may be applied to our stimuli, we need both to adopt the concepts to apply to stimuli in the LM contrast plane and to add a model component that handles contrast.

The angles used in our implementation of the LCM are angles in the LM contrast plane, with 0 corresponding to the positive abscissa, rather than angles in the CIELAB isoluminant plane. These angles are computed from the L- and M-cone contrasts of our stimulus modulations, as the arctangent of the ratio of M- to L-cone contrast:

$$\theta = tan^{-1}\left(\frac{m}{l}\right)$$

where $\theta$ is then the angle corresponding to one of our stimulus modulations.

Given the angular stimulus representation, each linear channel is characterized by its sensitivity to stimulation from each possible stimulus angle - basically a tuning function over angle. More specifically, the tuning functions are implemented as half-rectified cosines raised to an exponent. What differs across each channel is the phase of the tuning function (the peak sensitivity). Thus the sensitivity of the $i^{th}$ channel may be written as:

$$f_i(\theta) = \begin{cases} cos^n(\theta - \phi_i) & cos(\theta - \phi_i) \geq 0 \\ 0 & cos(\theta - \phi_i) < 0 \end{cases}$$

where $i$ indicates the channel, $n$ is the exponent which controls the narrowness of the channel tuning functions, and $\phi_i$ is the angle of peak sensitivity of the $i^{th}$ channel. In calculations, the angle $\theta$ is discretized, and both stimuli and mechanism tuning can be represented in matrix-vector form, as we describe below.

The first step of the model is to compute the channel responses. This is done with a vector representation of the stimulus. Each entry of the stimulus row vector represents one angle, and for the stimulus modulations we used, only one entry is non-zero for any given modulation. In our computations, we discretized angle in one degree steps (the resolution of the representation of $\theta$). The magnitude of the non-zero entry represents the contrast of the modulation. Similarly, the sensitivity of a channel may be represented by a column vector over the same angular discretization, with each entry of the vector being the sensitivity of the channel at the corresponding angle. The dot product of a stimulus vector with a channel vector yields the response of the channel to the stimulus.

The set of channels can be represented by the columns of a matrix C (nAngles x nChannels) and the set of stimuli represented by a matrix S (nStimuli x nAngles). Therefore, the hypothetical channel outputs, $H$ (nStimuli x nChannels), can be calculated as:

$$H = SC$$

The overall LCM response to a given stimulus is given as a weighted sum of the individual channel responses ($H$), with the weights, $w$ (nChannels x nVoxels) acting as parameters of the model. If we consider responses across a set of voxels, as was done by *Brouwer and Heeger, 2009*, the response $b$ (nStimuli x nVoxels) for a set of stimuli is given by:

$$b = Hw$$

In our analysis, we fit the LCM to the median time course of V1 and therefore we set $nVoxels = 1$.

In the context of the LCM, isoresponse contours are made up by the set of stimuli that satisfy the following:

$$Hw = SCw = k$$

where $k$ is a constant target response.

In their work, *Brouwer and Heeger, 2009* used six channels and an exponent of 2. The channels had peak sensitivities at 0°, 60°, 120°, 180°, 240° and 300°. Since our stimulus modulations were symmetric around around a background, we enforce that channels offset by 180° used the same weights in the linear combination. This results in three symmetric channels with pairs located at 0° and 180°, 60° and 240°, 120° and 300°.

One key difference between our experiment and the the original *Brouwer and Heeger, 2009* work is that we varied the stimulus contrast in each chromatic direction. To extend the LCM to handle contrast, we treated the overall LCM response as an equivalent contrast and passed this through a common Naka-Rushton function. This approach mirrors how we handled contrast in the the QCM. To model the BOLD fMRI response, we convolve the output of the Naka-Rushton with the HRF. We fit the LCM to our data using a method analogous to the one used to fit the QCM. More specifically, we found the three linear channel weights and the Naka-Rushton function parameters that yielded the best fit to the BOLD response time course. These parameters were found simultaneously through the use of MATLAB's fmincon optimization routine. We also fit a variant of the LCM analogous to a model used by *Kim et al., 2020* with sharper ($cos^6$) channel tuning and using eight rather than six underlying mechanisms.

Both versions of the LCM yield isoresponse contours similar in shape to those obtained with the QCM (*Figure 6—figure supplement 1*), with cross-validated $R^2$ values essentially the same as those obtained using the QCM (*Figure 4—figure supplement 1*). Notably, that the version of the LCM with more channels yielded an isoresponse contour that more closely approximated the isoresponse contour of the QCM than did the version of the LCM with fewer channels.

A key difference between the LCM and the QCM is in how the functional properties of the model relate to the L- and M-cone contrasts that characterize the early visual system responses to the stimuli. As discussed in the main text, the QCM can be implemented as the sum of the squared responses of two mechanisms, each of which responds as weighted sum of L- and M-cone contrast. Thus underlying the QCM is a quadratic (squaring) non-linearity. In the LCM, on the other hand, the underlying linear channels are tuned for angle in the LM contrast plane. Since angle is obtained as the arctangent of the ratio of M- and L-cone contrasts, the LCM is based on a different form of non-linearity than the QCM, which would entail different neural computations. To put it another way, the apparently simple linear form of the LCM when expressed in terms of stimulus angle is less simple when referred back to the L- and M-cone contrast representation. This same basic point applies to LCMs formulated with respect to hue angle in the CIELAB isoluminant plane, as in *Brouwer and Heeger, 2009* and *Kim et al., 2020*. Measurements of the response to mixtures of modulations might be used in the future to differentiate between the non-linearities embodied by the LCM and the QCM.

