## [Decision Letter]

**Acceptance summary:**

A key step towards understanding the cortical representation of color is to develop mathematical models that accurately and parsimoniously describe cortical representations of broad families of colorful stimuli. This study achieves this goal with the quadratic color model, a new benchmark f-or the quantitative assessment of cortical light responses that provides new insights into the underlying mechanisms.

**Decision letter after peer review:**

Thank you for submitting your article "A Quadratic Model Captures the Human V1 Response to Variations in Chromatic Direction and Contrast" for consideration by *eLife*. Your article has been reviewed by 3 peer reviewers, including Gregory D Horwitz as the Reviewing Editor and Reviewer #1, and the evaluation has been overseen by Tirin Moore as the Senior Editor. The following individual involved in review of your submission has agreed to reveal their identity: Stephen A Engel (Reviewer #2).

Essential revisions:

(1) The paper would be strengthened by a more thorough analysis of the QCM: comparison of the QCM to models more biologically plausible than the GLM, residuals analyzed for meaningful patterns, and quality-of-fit metrics compared with those of previous studies.

(2) The new results should be discussed in the context of previous EEG and fMRI studies.

(3) Trends in the data should be compared with predictions from psychophysics.

*Reviewer #1 (Recommendations for the authors):*

The quality of QCM fits could be appraised more rigorously. Are there any consistent patterns in the residuals? If so, what is the nature of the systematic deviations from the QCM, how big are they, and what might they indicate about the underlying biology?

The GLM allows jagged and non-monotonic contrast-response functions, which is unrealistic. A more useful benchmark would be a model in which contrast-response functions have a Naka-Rushton form, like they do in the QCM.

Neurons in the LGN respond more strongly to 12 Hz, full-field modulations than do most neurons in V1. One possibility is that much of the BOLD signal recorded from V1 under the conditions used in this study reflects the activity of LGN afferents. Some comments on this and potential consequences for the activation of V1 and beyond would be useful.

The changes in minor axis ratio and ellipse orientation with eccentricity are modest, but it is unclear how large we should expect them to be. Predictions from psychophysics, neurophysiology, or anatomy would provide a more useful benchmark than estimates of measurement error ("vertical spread of values at each eccentricity" and "set-to-set differences in parameters values").

More information should be giving regarding the bootstrap calculation of the 68% confidence intervals. Was the percentile method used? Was a normal distribution of parameter estimates assumed ({plus minus}1 SD of bootstrap resamples)?

Line 171: The measured BOLD percent signal change is described as "black line" in the text but more accurately as "thin gray line" in the figure legend.

Lines 626 and 627: Is an experimental run distinct from a functional run?

The discussions of cone fundamentals changing with eccentricity and contrast-response functions changing with eccentricity might be easier to understand if they were separated (both are described in the paragraph spanning lines 422-459).

The legends to Figure 12 and Supplementary Figure 9 are missing at least one line of text.

Typos: "Varience", "grayoordinate", "2-dimentional", "…and the bottom set of the show data for…"

*Reviewer #2 (Recommendations for the authors):*

This paper presents an important data set and model, and the successes of the paper are laid out in the other portions of this review process. Here I detail specific concerns that could be addressed in revision. Larger ones fall into four categories

1. Response patterns could be characterized even more completely, aiding readers.

Line 175: How does this R^2^ of the GLM compare to those reported in prior work? It would be good to know if it is in the standard range, which I bet it is.

Figure 2: Potting more of the results in terms of percent signal change rather than arbitrary units would allow comparison with previous results. This might also help readers interpret the lack of saturation of response as a function of contrast.

Line 281: It would be good to give readers a better sense of what responses are like outside of V1. Specifically, what quantitative estimates are there to support the claim that the stimuli are not driving activity beyond V1, even in V4 or V01? One solution might be to show a map of GLM variance explained throughout cortex, with some sample timecourses shown for V4ish and V01ish ROIs. It also may not be clear to non-specialists that neurons beyond V1 require spatial contrast to drive response; this could probably be highlighted more.

2. Additional comparisons of GLM and QCM.

General: While agreement between GLM and QCM is good, are there small but systematic differences? It might be helpful to include some sort of plot of residuals of the QCM fits. I think that is part of the point of Figure 7, but one could plot residuals explicitly, and it is not clear from the figure (at least to me) whether this plot is evaluating fits of just the contrast non-linearity or the entire QCM model. Figure 8: Another additional analysis that might be useful is a spatial map of a GLM to QCM comparison. It seems possible, at least in principle, that there are parts of the brain where the difference between the two models could be larger than in V1.

3. Discussion of other work.

While this is generally quite thorough, the manuscript would be strengthened by discussing the two additions mentioned in the public review: EEG results (e.g.Baseler and Sutter), and Brouwer and Heeger's "channel" model of color responses in cortex.

4. Discussion of implications for neurons in V1. A few additions to the discussion might be helpful. First, could the present results be dominated by input to V1 rather than action potentials generated within V1? This merits some discussion. Relatedly, readers might like to know if the present data set covered the LGN, which could be used in future work to address this issue.

Second, as mentioned above, additional discussion of what single unit work says about what kind of neurons will give strong responses to spatially uniform stimuli, and where they are located in cortex, could be useful.

Finally, and most trickily I suppose, is what conclusions about neurons can one make from the good QCM fit itself (besides lack of eccentricity effects). Can one conclude that for these stimulus conditions neurons in V1 whose preferred color direction is at or near L-M have higher gain than neurons whose preferred color direction is at or near L+M? I realize that there are a lot of assumptions (e.g. about separability and pooling of the signals by the fMRI response) being made in such a statement. I guess one could also build an "LGN-type" model that assumes all signal comes from just L+M or L-M tuned neurons, which could fit comparably to the QCM, but potentially with fewer parameters, since it assumes a 45 degree orientation of one population and a 135 degree orientation of another (almost equivalently a 2 channel version of the Brouwer and Heeger model). This is hinted at in line 370 in terms of psychophysics, but the model can be constrained by single unit data, at least from LGN. Is there anything useful one could conclude from such fits?

A final possible form this discussion could take is to tackle the question: Are there plausible neural response patterns that would be expected to not be well-fit by the QCM, and so the present data discomfirm them? I understand the authors' desire to not be speculative, but readers may already be speculating.

*Reviewer #3 (Recommendations for the authors):*

I think the main weakness here is the link between biology and data. Is it possible to implement a simple but biologically plausible population model of color processing in V1 (involving just the low SF-sensitive neurons) and test it against the data here? Such a model might have more than 6 parameters.… but fewer than 40: as the authors point out, the data are consistent with a model that sums approximately independent inputs from L-M and L+M channels. The authors might then discuss (qualitatively) how they expect the response functions to change as other populations of chromatically-sensitive neurons are excited by, for example, the presence of spatial structure.

Psychophysical data would also benefit the paper. I know there would be a lot of conditions – but with efficient staircases, data could be collected on at least some subjects to examine, for example, the observation that model parameters do not change across eccentricity (a surprising and interesting result!). Absent that, some more quantitative comparisons with existing psychophysical data (and neuroimaging data – for example Brouwer and Heeger's work) would be nice.

---

## [Author Response]

Reviewer #1 (Recommendations for the authors):The quality of QCM fits could be appraised more rigorously. Are there any consistent patterns in the residuals? If so, what is the nature of the systematic deviations from the QCM, how big are they, and what might they indicate about the underlying biology?

We agree with Reviewers 1 and 2 that an examination of model residuals could reveal systematic deviations from, and thus limitations of, the QCM. To address this, we examined the residuals for both the GLM and the QCM for each subject and session. We approached this as follows:

First, we analyzed the residuals as a function of stimulus condition, examining each direction/contrast pair separately. To do so, we plotted the residuals of the GLM and QCM fit to the BOLD signal time course over the 14 TRs after the start of each stimulus block. This produced a total of 10 residual curves per model and direction/contrast pair. We did not observe in this analysis any systematic variation in the residuals as a function of contrast level within a single chromatic direction.

Given this observation, and as a way to summarize the data, we examined the mean residual value, taken from 4 to 14 TRs after stimulus onset, for all trials in a color direction (collapsed over contrast). To check for systematic deviations in fit, we plotted these mean residuals for both the GLM and the QCM, separated by chromatic direction, for each subject and session. We examined these plots for consistent patterns. We observe no such patterns within or across models. Note in particular that the GLM has separate parameters for each stimulus direction and that the residual values for the GLM and QCM mostly overlap. The figure has been added to the manuscript as Figure 3 – supplemental figure 3 and the main text (section “Comparison of GLM and QCM”, starting on Line 227 of the main text) has been updated to convey this point.

The GLM allows jagged and non-monotonic contrast-response functions, which is unrealistic. A more useful benchmark would be a model in which contrast-response functions have a Naka-Rushton form, like they do in the QCM.

The reviewer brings up the good point that the non-monotonicity of the GLM fit to contrast response makes it likely on prior grounds that the GLM is overfitting the data. To provide a benchmark without this issue, we fit the data with models intermediate to the fully free GLM and the more constrained QCM. The most general of these models fit the data in each color direction with a separate Naka-Rushton function, allowing all but the offset parameters of this function to be independent across chromatic directions. More constrained versions of this model yoked additional Naka-Rushton parameters across directions. We explored locking the amplitude parameter (in addition to the offset), the exponent parameter (in addition to the offset), and yoking the amplitude, exponent, and offset parameters (allowing only the semi-saturation parameter to vary with chromatic direction).

To evaluate how well these models fit the data, we ran the same cross-validation procedure we used previously to compare the GLM to the QCM. Figure 4—figure supplement 1 shows this result for Subject 2 for both measurement sets (ignore the LCM BH, LCM Kim, and QCM locked bars in that plot for now; these will be described in response to other reviewer comments below). The magnitude of the cross-validated R^2^ for all of these Naka-Rushton models was only slightly higher than the GLM and not higher than the QCM. These results were similar for all subjects and measurement sets.

The primary conclusions from this analysis are that any overfitting by GLM is small and that the constraint across color directions imposed by the QCM does not decrease the cross-validated model fit relative to models that simply imposed by a smooth-monotonicity constraint on the contrast-response functions in each chromatic direction. We have updated the text in the section “Characterizing Cortical Responses with a Conventional GLM” (Line 157) to reflect these results. We also provide the cross-validation figures with the Naka-Rushton models for all subjects and measurement sets in the supplemental figures, replacing the previous cross-validation figures (Figure 4- supplemental figure 1). We left the cross-validation figure in the main text unchanged, so as not to take the more casual reader too far afield. For a similar reason, we retain the GLM as our baseline model.

Also note that we have removed the error bars from the cross-validation R^2^ values – we had computed these incorrectly in the initial submission. Although it would be possible to put error bars on these values by running the cross-validation analyses within a bootstrapping loop, this would be computationally intensive and we do not think doing so would provide additional useful information.

Neurons in the LGN respond more strongly to 12 Hz, full-field modulations than do most neurons in V1. One possibility is that much of the BOLD signal recorded from V1 under the conditions used in this study reflects the activity of LGN afferents. Some comments on this and potential consequences for the activation of V1 and beyond would be useful.

We agree. Our data do not distinguish the extent to which the response properties of the BOLD signals we measure in V1 are inherited from the LGN or are shaped by processing within V1.

To address this point, we have revised the discussion. We now point out that, even though the signals measured in our experiment are spatially localized to V1, we cannot ascribe the observed response properties to particular neural processing sites. As such, the V1 sensitivities found from fitting the QCM may be inherited from areas prior to V1. In principle, they could also be affected by feedback from other cortical areas. This text starts on Line 511 of the paper.

We also undertook an analysis of signals from the LGN in our data. Specifically, we interrogated the BOLD fMRI signal within individual subject subcortical segmentations defined using Freesurfer 7 and then registered to CIFTI space. Using this, we were able to extract a median time course for the LGN and fit the QCM in the same manner that we fit the time course for V1. For this analysis, the data from the two measurement sets within a subject were aggregated to increase signal-to-noise. Even so, data from two of the three subjects were too noisy to reveal reliable stimulus-driven responses. In one subject we were able to fit the QCM to data from the LGN, and found a similar minor axis ratio and ellipse angle as we found in V1. Yet even for this subject, responses were noisy, with a response amplitude approximately half of what we found in V1. Overall, we do not believe that the LGN responses are sufficiently well measured in our data for us to include these observations in the paper.

The changes in minor axis ratio and ellipse orientation with eccentricity are modest, but it is unclear how large we should expect them to be. Predictions from psychophysics, neurophysiology, or anatomy would provide a more useful benchmark than estimates of measurement error ("vertical spread of values at each eccentricity" and "set-to-set differences in parameters values").

In the original submission, we presented variation in QCM parameters across measurement sets and across voxels at similar eccentricities. These help estimate measurement precision: changes in parameter values that are consistently outside these ranges would point towards *significant* effects with eccentricity. We agree, however, that the original treatment did not address what might be a *meaningful* change with eccentricity. Consideration of the latter question is worthwhile, and in preparing this revision we have added an analysis and reworked the discussion to better address this question and to better set our measurements in the context of the prior literature. The relevant section of the discussion is “Change of Chromatic Sensitivity with Eccentricity” and begins on Line 531 of the paper.

The new analysis, shown in Figure 9 – supplemental figure 2 (see Results, Lines 350-363), uses the voxel-wise QCM fits to evaluate how L-M sensitivity and L+M sensitivity vary with eccentricity, a form of the data similar to that provided in earlier fMRI studies of V1. This analysis allows fairly direct comparison to what we think are the most directly relevant prior studies, those that used fMRI in V1 to examine this same question. Our results are in fairly striking contrast with the data of Mullen et al. (2007; their Figure 8) and Vanni et al. (2006; also their Figure 8), which we could take as measurements that define the magnitude of a meaningful effect of eccentricity. Summarizing their experiments and the size of their effects is not easy to do succinctly, so we simply refer the reader to the relevant figures in these two papers and note that if effects of that magnitude had been present in our data, they would have been apparent in our new analysis. At the same time, we note our data are consistent with those from a third prior study, that of D’Souza et al. (2016), who found little or no change in relative L-M and isochromatic sensitivity with increasing eccentricity in V1. We discuss possible reasons for these differences in results across studies (stimulus differences and some methodological issues), but a reconciliation of all of the studies is not currently in hand.

Prior psychophysical studies tell a fairly consistent story of a decline in L-M sensitivity relative to L+M sensitivity with increasing eccentricity, and we now acknowledge these studies more fully (see text starting at Line 547). We think direct comparison with psychophysics is tenuous because in general the patterns of BOLD fMRI responses in V1 do not always mirror effects found using psychophysical measurements of sensitivity (see section “Relation to Psychophysics in Discussion (Line 424)), so we don’t make much of this beyond noting it.

More information should be giving regarding the bootstrap calculation of the 68% confidence intervals. Was the percentile method used? Was a normal distribution of parameter estimates assumed ({plus minus}1 SD of bootstrap resamples)?

Indeed, we used percentiles to estimate error. We used 68% confidence intervals since this approximates +/- 1 SEM for a normal distribution. We have updated the text in section “Error Bar Generation” to make this clear.

Line 171: The measured BOLD percent signal change is described as "black line" in the text but more accurately as "thin gray line" in the figure legend.

Fixed.

Lines 626 and 627: Is an experimental run distinct from a functional run?

Good catch. These are the same thing and the wording has been unified in the revision.

The discussions of cone fundamentals changing with eccentricity and contrast-response functions changing with eccentricity might be easier to understand if they were separated (both are described in the paragraph spanning lines 422-459).

We have updated the text in an attempt to better explain these concepts, and made the points noted here in separate paragraphs (now starting on Line 584).

The legends to Figure 12 and Supplementary Figure 9 are missing at least one line of text.Typos: "Varience", "grayoordinate", "2-dimentional", "…and the bottom set of the show data for…"

Fixed.

Reviewer #2 (Recommendations for the authors):This paper presents an important data set and model, and the successes of the paper are laid out in the other portions of this review process. Here I detail specific concerns that could be addressed in revision. Larger ones fall into four categories.1. Response patterns could be characterized even more completely, aiding readers.Line 175: How does this R^2^ of the GLM compare to those reported in prior work? It would be good to know if it is in the standard range, which I bet it is.

We gave this some thought, and went looking for comparable studies to offer, but ultimately decided that this quantitative comparison is difficult to properly frame. Comparison of R^2^ values across studies is tricky, given that variance explained will be affected not only by how well the model accounts for the underlying measurements, but also by other factors that affect SNR and that may differ across studies. A particular challenge here is that we fit the concatenated (as opposed to average) time-series data across multiple acquisitions and sessions. As a consequence, the R^2^ values will be lower than in those studies that have repeated presentations of the same stimulus sequence and thus have performed signal averaging prior to model fitting and subsequent report of R^2^ values. For these reasons we don’t have a particular comparison set of R^2^ values to provide from other studies. We continue to feel, however, that expressing the quality of our model fit in terms of R^2^ provides a useful metric for comparison across subjects and runs within our study. We now mention these points (see Lines 404).

Figure 2: Potting more of the results in terms of percent signal change rather than arbitrary units would allow comparison with previous results. This might also help readers interpret the lack of saturation of response as a function of contrast.

We agree that presenting work in a way that is interpretable in context of other studies is important. To accomplish this goal, we provide plots of the BOLD fMRI time course, in units of percent signal change, together with the model predictions.

Line 281: It would be good to give readers a better sense of what responses are like outside of V1. Specifically, what quantitative estimates are there to support the claim that the stimuli are not driving activity beyond V1, even in V4 or V01? One solution might be to show a map of GLM variance explained throughout cortex, with some sample timecourses shown for V4ish and V01ish ROIs. It also may not be clear to non-specialists that neurons beyond V1 require spatial contrast to drive response; this could probably be highlighted more.

To address this, we have fit the GLM vertex-wise to the time series data from the whole brain. We generally found that the variance explained in vertices outside of V1 was close to zero, with the exception of a small patch of values in the vicinity of V4/VO1. The GLM variance explained in this area was roughly half of that explained within V1. To further examine these regions, we used the subject-specific registration of the retinotopic atlas from Wang et al., as implemented by Neuropythy. We then transferred the V4 and the VO1 ROIs into HCP CIFTI space and extracted the median time series for each ROI. With these median time series, we fit the QCM in V4 and VO1 for all subjects and measurement sets. The parameters of the QCM fit for hV4 and VO1 were generally consistent with those of V1, although fit quality was overall worse. While examining signals in downstream areas is an exciting research direction, as with our measurements of the LGN, we feel that the data quality prevents us from making concrete statements about the response in these areas in this paper. We provide the average variance explained map for early visual cortex (EVC, the spatial extent of the Benson template) for the GLM. The spatial extent of these maps includes hV4 and VO1. We have added these average EVC maps as Figure 8 figure supplemental 1, and indicate in section “Isoresponse Contour Parameter Maps” (Line 300) that the responses outside of V1 were not sufficiently reliable for us to report model fits to those responses.

2. Additional comparisons of GLM and QCM.General: While agreement between GLM and QCM is good, are there small but systematic differences? It might be helpful to include some sort of plot of residuals of the QCM fits. I think that is part of the point of Figure 7, but one could plot residuals explicitly, and it is not clear from the figure (at least to me) whether this plot is evaluating fits of just the contrast non-linearity or the entire QCM model.

We agree. We have addressed this as described in our response to a similar suggestion from Reviewer 1, above.

Figure 8: Another additional analysis that might be useful is a spatial map of a GLM to QCM comparison. It seems possible, at least in principle, that there are parts of the brain where the difference between the two models could be larger than in V1.

We agree that this is a good check. We took the difference between the GLM and the QCM variance explained maps, averaged across subjects. This difference map is provided Author response image 1. We did not observe any large deviations in fit in the EVC region of interest. The map contains only positive values showing that the GLM had higher R^2^ values, as expected since the QCM is a parametrized subset of the GLM. The largest difference in variance explained between the maps is 0.03, demonstrating nearly equivalent performance in non-cross-validated model fit. We did not add this figure, although we could if the reviewers or and editor feel it important to do so; we do now, however, provide both of the variance explained maps used to produce this figure (Figure 8 (bottom row) and Figure 8 figure supplemental 1).

3. Discussion of other work.While this is generally quite thorough, the manuscript would be strengthened by discussing the two additions mentioned in the public review: EEG results (e.g.Baseler and Sutter), and Brouwer and Heeger's "channel" model of color responses in cortex.

We thank the reviewer for pointing out these papers for comment. We now cite the Baseler and Sutter paper in the section “Change of Chromatic Sensitivity with Retinal Eccentricity” as an example of a prior measurement of variation in chromatic sensitivity with eccentricity. We now also treat the Brouwer and Heeger channel model; this treatment is described fully in the response to Reviewer 3 below.

4. Discussion of implications for neurons in V1. A few additions to the discussion might be helpful. First, could the present results be dominated by input to V1 rather than action potentials generated within V1? This merits some discussion. Relatedly, readers might like to know if the present data set covered the LGN, which could be used in future work to address this issue.

Indeed, the BOLD signals we measured in V1 may inherit their response properties from LGN inputs. Reviewer 1 made a similar comment, please see response above.

Second, as mentioned above, additional discussion of what single unit work says about what kind of neurons will give strong responses to spatially uniform stimuli, and where they are located in cortex, could be useful.

We have added text in the discussion to the section now titled “Relation to Underlying Mechanisms” to discuss briefly the relationship between our stimuli and responses of known cell types. See text in paragraph starting on Line 490.

Finally, and most trickily I suppose, is what conclusions about neurons can one make from the good QCM fit itself (besides lack of eccentricity effects). Can one conclude that for these stimulus conditions neurons in V1 whose preferred color direction is at or near L-M have higher gain than neurons whose preferred color direction is at or near L+M? I realize that there are a lot of assumptions (e.g. about separability and pooling of the signals by the fMRI response) being made in such a statement. I guess one could also build an "LGN-type" model that assumes all signal comes from just L+M or L-M tuned neurons, which could fit comparably to the QCM, but potentially with fewer parameters, since it assumes a 45 degree orientation of one population and a 135 degree orientation of another (almost equivalently a 2 channel version of the Brouwer and Heeger model). This is hinted at in line 370 in terms of psychophysics, but the model can be constrained by single unit data, at least from LGN. Is there anything useful one could conclude from such fits?

As with the comment just above, we have revised the discussion to be more elaborated on this point. See paragraphs starting at Line 490 down to the end of that section of the discussion.

A final possible form this discussion could take is to tackle the question: Are there plausible neural response patterns that would be expected to not be well-fit by the QCM, and so the present data discomfirm them? I understand the authors' desire to not be speculative, but readers may already be speculating.

To address this general point, we added an analysis to show that there are parametric isoresponse contour shapes that our data have sufficient power to reject. To do this, we fit and cross-validated a form of the QCM with the angle constrained to by 0 degrees. See the bar plot provided in Figure 4—figure supplement 1, cross-validated R^2^ labeled QCM locked. The lower cross-validated R^2^ for this model makes the point, and that R^2^ is now provided as part of Figure 4 figure supplement 1. We treat this in the discussion in paragraph starting at Line 481

Reviewer #3 (Recommendations for the authors):I think the main weakness here is the link between biology and data. Is it possible to implement a simple but biologically plausible population model of color processing in V1 (involving just the low SF-sensitive neurons) and test it against the data here? Such a model might have more than 6 parameters.… but fewer than 40: as the authors point out, the data are consistent with a model that sums approximately independent inputs from L-M and L+M channels. The authors might then discuss (qualitatively) how they expect the response functions to change as other populations of chromatically-sensitive neurons are excited by, for example, the presence of spatial structure.

To our knowledge, the only biologically plausible population model of V1 color processing that incorporates spatial frequency information in the prediction of the BOLD response is the model by Schluppeck and Engel (2002). Their model is based on the primate electrophysiological data of Johnson et al. (2001) and from this, produces isoresponse contours which were compared to BOLD fMRI isoresponse contours. The Schluppeck and Engel model at low spatial frequencies has spatial sensitivity differences for ‘color’, ‘color-luminance’ and ‘luminance’ cells roughly on the order of the average minor axis ratio in our study. Further, given that we have no spatial frequency variations, any variations in model output for our stimuli are driven by the color stage of the model. Schluppeck and Engel showed that the shapes of their predicted isoresponse contours were elliptical and matched those of Engel et al. 1997. Therefore, we believe that their model is consistent with the QCM. Our revised discussion of the link to underlying mechanisms is now in section “Relation to Underlying Mechanisms” starting on Line 449. We discuss some ideas about how one might proceed to think about incorporating spatial structure into measurements and models in the last paragraph of the paper, starting at Line 638.

We take the point that the high number of parameters of the GLM are highly unlikely to correspond to an equally high number of underlying mechanisms in V1, and we do not mean to imply this. We use the GLM as a comparison point that allows each stimulus condition to have an individual response weighting independent of the other conditions which will provide the best fit to the data, of the models tested. As an approach to the question of what happens as one adds more parameters than the QCM has, but fewer than the GLM, we now fit our data with two variants of the Brouwer and Heeger (2009) model, one having the same number of parameters as the QCM, and a variant of this model as implemented by Kim et al. (2020), which has more parameters. This exercise is described in more detail just below.

Psychophysical data would also benefit the paper. I know there would be a lot of conditions – but with efficient staircases, data could be collected on at least some subjects to examine, for example, the observation that model parameters do not change across eccentricity (a surprising and interesting result!). Absent that, some more quantitative comparisons with existing psychophysical data (and neuroimaging data – for example Brouwer and Heeger's work) would be nice.

We have addressed this comment in two ways. First, we now discuss the psychophysical literature that argues that very careful measurements of isothreshold contours do not reject the hypothesis that these contours are elliptical (section now called “Relation to Psychophysics”, Line 424). Second, as described above we have elaborated our discussion of the effects of eccentricity.

In terms of fitting the QCM to other datasets. A key feature of the QCM is the fact that it provides an account of how response varies with both chromatic direction and contrast, and indeed estimating isoresponse contours requires either using some sort of procedure that equates response across chromatic directions or (as we have done) varies both chromatic direction and contrast. The Brouwer and Heeger dataset, for example, uses only one contrast per chromatic direction and leverages the variation of response with chromatic direction in its decoding analysis. As a consequence, the Brouwer and Heeger dataset for example would not be appropriate for testing the QCM or more generally any model that describes the shape of the isoresponse contour.

Reviewer 2 suggests that comparison to a model intermediate to the GLM and QCM would be of interest. To address this, we have added additional model comparisons to the paper. As suggested, we used the work of Brouwer and Heeger as a point of departure. They developed a model based on mechanisms tuned according to cos^2^ functions of angle in the isoluminant plane, generated as the squares of linear response mechanisms in the CIELAB color space. They then modelled data collected at a single contrast using a linear combination of the responses of these mechanisms, which implies an isorespone contour. Rather than allowing mechanism tuning to vary, they varied the weights with which the mechanism outputs were combined.

To apply these concepts to our dataset, we formed mechanisms as cos^2^ functions of angle in the LM contrast plane, and combined these linearly to define isocontrast contours. Since our stimuli are symmetric modulations around a white point, we used three symmetric rather than six independent mechanisms. To handle the variations in contrast, we passed the output of the linearly summed mechanisms through a common Naka-Rushton function. We call this the Linear Channels Model (LCM). As we now show in the supplement, this approach accounts for our data about as well as the QCM, and produces a similar isoresponse contour, although the detailed shape varies from that of the ellipse.

More generally, any model that can describe an isoresponse contour of the general shape we observe is likely to fit the data equally well. We also fit a variant of the LCM with sharper tuning and more parameters (Kim et al. 2020) and show that this again leads to similar fit quality and isoresponse contours. We fit this model to all subjects and measurement sets. Figure 6—figure supplement 1 shows the isoresponse contours derived from the LCM for both the Brouwer and Heeger model and the Kim et al. model plotted with the QCM ellipse for Subject 2. These contours are similar, with the LCM contours having small wiggles around the ellipse. Notably, with more mechanisms, the LCM contours approach the form of an ellipse, and not more convoluted shapes. To assess how well this model performed, we employed the same cross-validation routine used to compare all other models. The results of this can be seen in the cross-validated R^2^ bar plot in Figure 4—figure supplement 1. This plot compares the cross-validated variance explained by the GLM, the Naka-Rushton variants, the QCM, and both versions of the LCM. There is very little difference in cross-validated R^2^ between the QCM and the two variants of the LCM, with the GLM just slightly worse.

Thus our data tell us the basic shape of the underlying isoresponse contours, but there are limits to the inferences we can make about the underlying mechanisms that produce the contours. We present these ideas in detail in the Appendix 1, and refer to them in the Discussion of the main paper (section “Relation to Underlying Mechanisms”, Line 449, Figure 6—figure supplement 1). The Appendix 1 also provides some brief comments on differences in the required mechanistic operations required to implement the QCM and LCM.